# $\mathcal{G}$-MIXUP: GRAPH DATA AUGMENTATION FOR GRAPH CLASSIFICATION

## ABSTRACT

This work develops *mixup to graph data*. Mixup has shown superiority in improving the generalization and robustness of neural networks by interpolating features and labels of random two samples. Traditionally, Mixup can operate on regular, grid-like, and Euclidean data such as image or tabular data. However, it is challenging to directly adopt Mixup to augment graph data because two graphs typically: 1) have different numbers of nodes; 2) are not readily aligned; and 3) have unique topologies in non-Euclidean space. To this end, we propose $\mathcal{G}$-Mixup to augment graphs for graph classification by interpolating the generator (i.e., graphon) of different classes of graphs. Specifically, we first use graphs within the same class to estimate a graphon. Then, instead of directly manipulating graphs, we interpolate graphons of different classes in the Euclidean space to get mixed graphons, where the synthetic graphs are generated through sampling based on the mixed graphons. Extensive experiments show that $\mathcal{G}$-Mixup substantially improves the generalization and robustness of GNNs.

## 1 INTRODUCTION

Recently deep learning has been widely adopted to graph analysis. Graph Neural Networks (GNNs) (Wu et al., 2020; Zhou et al., 2020b; Zhang et al., 2020; Xu et al., 2018) have made many significant breakthroughs on graph classification. Meanwhile, data augmentation (e.g., DropEdge (Rong et al., 2020), Subgraph (You et al., 2020)) has also been adopted to graph analysis by generating synthetic graphs to create more training data for improving the generalization of graph classification models. However, existing graph data augmentation strategies typically aim to augment graphs at a *within-graph* level by either modifying edges or nodes in individual graph, which limits them to only generating new graphs based on one individual graph. The *between-graph* augmentation methods (i.e., augmenting graphs between graphs) are still under-explored.

In parallel with the development of graph neural networks, Mixup (Zhang et al., 2017) and its variants (e.g., Manifold Mixup (Verma et al., 2019)), as data augmentation methods, have been theoretically and empirically shown to improve the generalization and robustness of deep neural networks in image recognition (Zhang et al., 2017; Verma et al., 2019; Zhang et al., 2021) and natural language processing (Guo et al., 2019; Guo, 2020). The basic idea of Mixup is to linearly interpolate continuous values of random sample pairs to generate more synthetic training data. The formal mathematical expression of Mixup is $\mathbf{x}_{new} = \lambda \mathbf{x}_i + (1 - \lambda)\mathbf{x}_j, \mathbf{y}_{new} = \lambda \mathbf{y}_i + (1 - \lambda)\mathbf{y}_j$, where $(\mathbf{x}_i, \mathbf{y}_i)$ and $(\mathbf{x}_j, \mathbf{y}_j)$ are two samples drawn at random from training data and the target $\mathbf{y}$ are one-hot labels. With *graph neural network* and *mixup* in mind, the following question naturally arises,

> ***Can we mix up graph data to improve the generalization and robustness of GNNs?***

It remains an open and challenging problem to mix up the graph data due to the characteristics of graphs and the requirements of applying Mixup. Typically, Mixup requires that original data instances are regular and well-aligned in Euclidean space, such as image data and table data. However, graph data is distinctly different from image data due to the following characteristics: *(i) graph data is irregular*. The number of nodes in different graphs are typically different to others; *(ii) graph data is not well-aligned*. The nodes in different graphs can not be aligned well directly; *(iii) graph topology between classes are divergent*. The topologies of a pair of graphs from different classes are usually different while the topologies of those from the same class are usually similar. thus make it challenging to directly adopt the Mixup strategy to graph data.

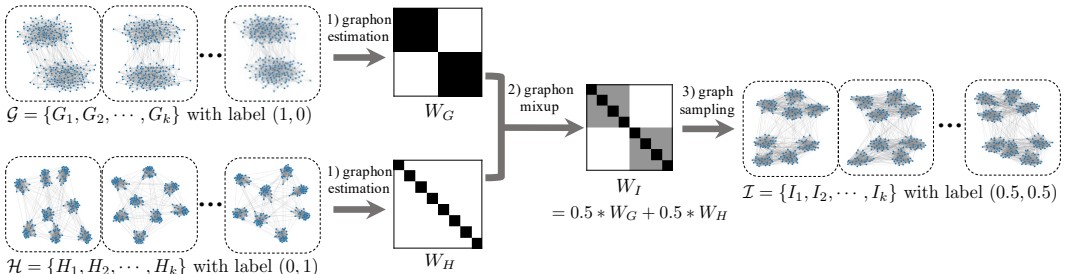

Figure 1: An overview of $\mathcal{G}$-Mixup. The task is binary graph classification. We have two classes of graphs $\mathcal{G}$ and $\mathcal{H}$ with different topologies ($\mathcal{G}$ has two communities while $\mathcal{H}$ has eight communities). $\mathcal{G}$ and $\mathcal{H}$ have different graphons. We mix up the graphons $W_{\mathcal{G}}$ and $W_{\mathcal{H}}$ to obtain a mixed graphon $W_{\mathcal{I}}$, and then sample new graphs from the mixed graphon. Intuitively, the synthetic graphs have two major communities and each of which has four sub-communities, demonstrating that the generated graphs preserve the structure of original graphs from both classes.

To tackle the aforementioned problems, we propose $\mathcal{G}$-Mixup, a class-level graph data augmentation method, to mix up graph data based on graphons. The graphs within one class have the same generator (i.e., graphon). We mix up the graphons of different classes and then generate synthetic graphs. Informally, a graphon can be thought of as a probability matrix (e.g., the matrix $W_G$ and $W_H$ in Figure 1), where $W(i, j)$ represents the probability of edge between node $i$ and $j$. The real-world graphs can be regraded as generated from graphons. In this way, we can mix two classes of graphs by mixing their generators. Since the graphons of different graphs is regular, well-aligned, and is defined in Euclidean space, it is easy and natural to mix up the graphons and then generate the synthetic graphs therefrom. We also provide theoretical analysis of graphons mixup, which guarantees that the generated graphs will preserve the key characteristics of both original classes. Our proposed method is illustrated in Figure 1 with an example. Given two graph training sets $\mathcal{G} = \{G_1, G_2, \cdots, G_k\}$ and $\mathcal{H} = \{H_1, H_2, \cdots, H_k\}$ with different labels and distinct topologies (i.e., $\mathcal{G}$ has two communities while $\mathcal{H}$ has eight communities), we estimate graphons $W_{\mathcal{G}}$ and $W_{\mathcal{H}}$ respectively from $\mathcal{G}$ and $\mathcal{H}$. We then mix up the two graphons and obtain a mixed graphon $W_{\mathcal{I}}$. After that, we sample synthetic graphs from $W_{\mathcal{I}}$ as additional training graphs. The generated synthetic graphs have two major communities and each of them have four sub-communities, which is a mixture of the two graph sets. It thus shows that $\mathcal{G}$-Mixup is capable of mixing up graphs.

In summary, our **main contributions** are three-fold. *Firstly*, we propose $\mathcal{G}$-Mixup to augment the training graphs for graph classification. Since directly mixing up graphs is intractable, $\mathcal{G}$-Mixup mixes the graphons of different classes of graphs to generate synthetic graphs. *Secondly*, we theoretically prove that the synthetic graph will be the mixture of the original graphs, where the key topology (i.e., discriminative motif) of source graphs will be preserved. *Thirdly*, we demonstrate the efficacy of the proposed $\mathcal{G}$-Mixup on various graph neural network backbones and datasets. Extensive experimental results show the proposed $\mathcal{G}$-Mixup substantially improves the performance of graph neural networks in terms of enhancing their generalization and robustness.

## 2 PRELIMINARIES

In this section, we first go over the notations used in this paper, and then introduce graph related concepts including graph homomorphism and graphons, which will be used for theoretical analysis in this work. Finally, we briefly review the graph neural networks for graph classification.

### 2.1 NOTATIONS

Given a graph $G$, we use $V(G)$ and $E(G)$ to denote its nodes and edges, respectively. The number of nodes is $\mathrm{v}(G) = |V(G)|$, and the number of edges is $\mathrm{e}(G) = |E(G)|$. We use $G, H, I$ to denote graphs and $\mathcal{G}, \mathcal{H}, \mathcal{I}$ to denote graph set. $\mathbf{y}_{\mathcal{G}} \in \mathbb{R}^C$ denotes the label of graph set $\mathcal{G}$, where $C$ is number of classes of graphs. A graph could contain some interesting and frequent patterns and subgraphs which are called *motifs*. The motifs in graph $G$ is denoted as $F_G$. The set of motifs in graph set $\mathcal{G}$ is denoted as $\mathcal{F}_{\mathcal{G}}$. $W_{\mathcal{G}}$ denotes the graphon of graph set $\mathcal{G}$. $\mathbf{W}$ denotes the step

function. $\text{Unif}_{[0,1]}$ denotes the uniform distribution between 0 and 1. $\text{Bern}(\cdot)$ denotes the Bernoulli distribution. $\mathbb{G}(n, W)$ denotes the random graph with $n$ nodes based on graphon $W$.

## 2.2 GRAPH HOMOMORPHISM AND GRAPHONS

**Graph Homomorphism.** A graph homomorphism is an adjacency-preserving mapping between two graphs, i.e., mapping adjacent vertices in one graph to adjacent vertices in the other. Formally, a *graph homomorphism* $\phi\colon F \to G$ is a map from $V(F)$ to $V(G)$, where if $\{u, v\} \in E(F)$, then $\{\phi(u), \phi(v)\} \in E(G)$. For two graphs $H$ and $G$, there could be multiple graph homomorphisms between them. Let $\text{hom}(H, G)$ denotes the total number of graph homomorphisms from graph $H$ to graph $G$. For example, $\text{hom}(\bullet, G) = |V(G)|$ if graph $H$ is $\bullet$, $\text{hom}(\bullet\!\!-\!\!\bullet, G) = 2|E(G)|$ if graph $H$ is $\bullet\!\!-\!\!\bullet$, and $\text{hom}(\triangle, G)$ is six times the number of triangles in $G$. There are in total $|V(G)|^{|V(H)|}$ mappings from $H$ to $G$, but only some of them are homomorphisms. Thus, we define *homomorphism density* to measure the relative frequency that the graph $H$ appears in graph $G$:

$$t(H, G) = \frac{\text{hom}(H, G)}{|V(G)|^{|V(H)|}}.$$

For example, $t(\bullet, G) = |V(G)|/n^1 = 1$, $t(\bullet\!\!-\!\!\bullet, G) = 2|E(G)|/n^2$.

**Graphon.** A graphon (Airoldi et al., 2013) is a continuous, bounded and symmetric function $W : [0, 1]^2 \to [0, 1]$ which may be thought of as the weight matrix of a graph with infinite number of nodes. Then, given two points $u_i$, $u_j \in [0, 1]$, $W(i, j)$ represents the probability that nodes $i$ and $j$ be related with an edge. Various quantities of a graph can be calculated as a function of the graphon. For example, the degree of nodes in graphs can be easily extended to a degree distribution function in graphons, which is characterized by its graphon marginal $d_W(x) = \int_0^1 W(x, y)dy$. Similarly, the concept of homomorphism density can be naturally extended from graphs to graphons. Given an arbitrary graph motif $F$, its homomorphism density with respect to graphon $W$ is defined by

$$t(F, W) = \int_{[0,1]^{V(F)}} \prod_{i,j \in E(F)} W(x_i, x_j) \prod_{i \in V(F)} dx_i.$$

For example, the edge density of graphon $W$ is $t(\bullet\!\!-\!\!\bullet, W) = \int_{[0,1]^2} W(x, y)\, dxdy$, and the triangle density of graphon $W$ is $t(\triangle, W) = \int_{[0,1]^3} W(x, y)W(x, z)W(y, z)\, dxdydz$.

## 2.3 GRAPH CLASSIFICATION WITH GRAPH NEURAL NETWORKS

Given a set of graphs, graph classification aims to assign a class label for each graph $G$. Recently, graph neural networks have become the state-of-the-art approach for graph classification. Without loss of generalization, we present the formal expression of a graph convolution network (GCN) (Kipf & Welling, 2016). The forward propagation at $k$-th layer is described as the following:

$$\mathbf{a}_i^{(k)} = \text{AGG}^{(k)}\left(\left\{\mathbf{h}_j^{(k-1)} : j \in \mathcal{N}(i)\right\}\right), \quad \mathbf{h}_i^{(k)} = \text{COMBINE}^{(k)}\left(\mathbf{h}_i^{(k-1)}, \mathbf{a}_i^{(k)}\right), \quad (1)$$

where $\mathbf{h}_i^{(k)} \in \mathbb{R}^{n \times d_k}$ is the intermediate representation of node $i$ at the $k$-th layer, $\mathcal{N}(i)$ denotes the neighbors of node $i$. $\text{AGG}(\cdot)$ is an aggregation function to collect embedding representations from neighbors, and $\text{COMBINE}(\cdot)$ combines neighbor representation and its representation at $(k-1)$-th layer followed by nonlinear transformation. For graph classification, a graph-level representation is obtained by summarizing all node-level representations in the graph by a readout function:

$$\mathbf{h}_G = \text{READOUT}\left(\left\{\mathbf{h}_i^{(k)} : i \in E(G)\right\}\right), \quad \hat{\mathbf{y}} = \text{softmax}(\mathbf{h}_G), \quad (2)$$

where $\text{READOUT}(\cdot)$ is the readout function, which can be a simple function such as average or sophisticated pooling function (Gao & Ji, 2019; Ying et al., 2018), $\mathbf{h}_G$ is the representation of graph $G$, and $\hat{\mathbf{y}} \in \mathbb{R}^C$ is the output estimating the probability that $G$ belongs to each of the $C$ classes.

## 3 METHODOLOGY

In this section, we first formally introduce the proposed $\mathcal{G}$-Mixup, and then present theoretical analysis of graph generation via graphon interpolation from the graph homomorphism density perspective.

### 3.1 $\mathcal{G}$ -MIXUP

Different from interpolation of image data in Euclidean space, adopting Mixup to graph data is nontrivial since graphs are irregular, unaligned and non-Euclidean data. In this work, we show that this challenge could be tackled via graphon theories. By intuition, graphon can be thought of as a generator to generate graphs. The real-world graphs can be regraded as generated form a graphon, which has same homomorphism density of arbitrary motif to that of graphon. With this in mind, we propose $\mathcal{G}$-Mixup, a class-level data augmentation via graphon interpolation. $\mathcal{G}$-Mixup interpolates different graph generator to obtain a new generator. Then, synthetic graphs are sampled based on the mixed graphon for graph data augmentation. The graphs sampled from this generator partially possess properties of all the original graphs. Formally, $\mathcal{G}$-Mixup can be formulated as the following:

$$\text{Graphon Estimation:} \qquad \mathcal{G} \to W_{\mathcal{G}}, \mathcal{H} \to W_{\mathcal{H}}, \tag{3}$$

$$\text{Graphon Mixup:} \qquad W_{\mathcal{I}} = \lambda W_{\mathcal{G}} + (1 - \lambda)W_{\mathcal{H}}, \tag{4}$$

$$\text{Graph Generation:} \quad \{I_1, I_2, \cdots, I_k\} \overset{\text{i.i.d}}{\sim} \mathbb{G}(k, W_{\mathcal{I}}), \tag{5}$$

$$\text{Label Mixup:} \qquad \mathbf{y}_{\mathcal{I}} = \lambda \mathbf{y}_{\mathcal{G}} + (1 - \lambda)\mathbf{y}_{\mathcal{H}}, \tag{6}$$

where $W_{\mathcal{G}}, W_{\mathcal{H}}$ are graphons of the graph set $\mathcal{G}$ and $\mathcal{H}$. The mixed graphon is denoted by $W_{\mathcal{I}}$, and $\lambda \in [0, 1]$ is the trade-off hyperparameter to control the contributions from different original graphs for interpolation. The set of synthetic graphs generated from $W_{\mathcal{I}}$ is $\mathcal{I} = \{I_1, I_2, \cdots, I_k\}$. The $\mathbf{y}_{\mathcal{G}} \in \mathbb{R}^C$ and $\mathbf{y}_{\mathcal{H}} \in \mathbb{R}^C$ are vectors containing ground-truth labels for graph $G$ and $H$ respectively. The label vector of a generated graph in $\mathcal{I}$ is set as $\mathbf{y}_{\mathcal{I}} \in \mathbb{R}^C$, where $C$ is the total classes of graphs.

As illustrated in Figure 1 and the above equations, the proposed $\mathcal{G}$-Mixup includes three key steps: **i) estimate the graphon** for each class of graphs with the same label, **ii) mix up the graphons** of different classes of graphs, and **iii) generate the synthetic graphs** based on the mixed graphon. Specifically, we have $\mathcal{G} = \{G_1, G_2, \cdots, G_k\}$ with label $\mathbf{y}_G$, and $\mathcal{H} = \{H_1, H_2, \cdots, H_k\}$ with label $\mathbf{y}_{\mathcal{H}}$. Graphons $W_{\mathcal{G}}$ and $W_{\mathcal{H}}$ are estimated from graph set $\mathcal{G}$ and $\mathcal{H}$, then we mix them up though linearly interpolating two graphons and their training labels and obtain $W_{\mathcal{I}}$ and $\mathbf{y}_{\mathcal{I}}$. The synthetic graph set $\mathcal{I}$ is sampled based on $W_{\mathcal{I}}$, which will be used as additional training graphs.

### 3.2 IMPLEMENTATION

In this section, we introduce the details of the implementation of estimating graphon from the observed graphs and generating synthetic graphs in the real-world scenario.

**Graphon Estimation and Mixup.** Estimating graphons from observed graphs is a prerequisite for $\mathcal{G}$-Mixup, however, it is intractable because graphon is an unknown function without closed-form solution in real-world graphs. Therefore, we use step function (Lovász, 2012; Xu et al., 2021) as an approximation of graphon. [1] The step function estimation methods are well-studied, which first aligns the nodes of a series of graphs based on simple node measurements (e.g., degree) and then estimates the step function from all the aligned adjacency matrices. The step function estimation methods used includes sorting-and-smoothing (SAS) method (Chan & Airoldi, 2014), stochastic block approximation (SBA) (Airoldi et al., 2013), "largest gap" (LG) (Channarond et al., 2012), matrix completion (MC) (Keshavan et al., 2010), universal singular value thresholding (USVT) (Chatterjee et al., 2015). Formally, a step function $\mathbf{W}^P : [0, 1]^2 \mapsto [0, 1]$ is define as $\mathbf{W}^P(x, y) = \sum_{k, k'=1}^{K} w_{kk'} \mathbb{1}_{\mathcal{P}_k \times \mathcal{P}_{k'}}(x, y)$, where $\mathcal{P} = (\mathcal{P}_1, .., \mathcal{P}_K)$ denotes the partition of $[0, 1]$ into $K$ adjacent intervals of length $1/K$, $w_{kk'} \in [0, 1]$, and indicator function $\mathbb{1}_{\mathcal{P}_k \times \mathcal{P}_{k'}}(x, y)$ equals to 1 if $(x, y) \in \mathcal{P}_k \times \mathcal{P}_{k'}$ and otherwise it is 0. *Essentially, the step function can be seen as a matrix* $\mathbf{W} = [w_{kk'}] \in [0, 1]^{K \times K}$, *where $\mathbf{W}_{ij}$ is the probability of edge between node $i$ and $j$. In practice, we use the matrix-form step function as graphon to mix up and generation synthetic graphs.*

*For binary classification*, we have $\mathcal{G} = \{G_1, G_2, \cdots, G_k\}$ and $\mathcal{H} = \{H_1, H_2, \cdots, H_k\}$ with different labels, we estimate their step functions $\mathbf{W}_{\mathcal{G}} \in \mathbb{R}^{K \times K}$ and $\mathbf{W}_{\mathcal{H}} \in \mathbb{R}^{K \times K}$ We let $K$ be the average number of nodes in all graphs. *For multi-class classification*, we first estimate the step function for each class of graphs and then randomly select two to perform mix-up. The resultant step function is $\mathbf{W}_{\mathcal{I}} = \lambda \mathbf{W}_{\mathcal{G}} + (1 - \lambda)\mathbf{W}_{\mathcal{H}} \in \mathbb{R}^{K \times K}$, which serves as the generator of synthetic graphs.

---

[1]Because weak regularity lemma of graphon (Frieze & Kannan, 1999) indicates that an arbitrary graphon can be approximated well by step function. Detailed discussion is in Appendix A.4.

**Synthetic Graphs Generation.** A graphon $W$ provides a distribution to generating arbitrarily sized graphs. Specifically, a $k$-node random graph $\mathbb{G}(k, W_{\mathcal{I}})$ can be generated following the process:

$$u_1, \ldots, u_k \overset{\text{iid}}{\sim} \text{Unif}_{[0,1]}, \quad (\mathbb{G}(k, W)_{ij} | u_1, \ldots, u_k) \overset{\text{iid}}{\sim} \text{Bern}(W(u_i, u_j)), \forall i, j \in [k]. \qquad (7)$$

Since we only estimate the step function $\mathbf{W}$ to approximate the graph $W$, we set $W(u_i, u_j) = \mathbf{W}_I[\lfloor 1/u_i \rfloor, \lfloor 1/u_j \rfloor]$, and $\lfloor \cdot \rfloor$ is the floor function. The first step samples $K$ nodes independently from a uniform distribution on $[0, 1]$. The second step generates an adjacency matrix $\boldsymbol{A} = [a_{ij}] \in \{0, 1\}^{K \times K}$, whose element values follow the Bernoulli distributions determined by the step function. A graph is thus obtained as $G$ where $V(G) = \{1, ..., K\}$ and $E(G) = \{(i, j) \mid a_{ij} = 1\}$. A set of synthetic graphs can be generated by conducting the above process independently multiple times. For node features, we generate them of synthetic graphs based the original two sets of graphs. Specifically, we generate the node feature of each graphons. In the graphon estimation, we align the node features with the process of the adjacency matrix. For each graphon we have a set of node features, we can pooling the node features to obtain the *graphon features*. The the node features would inherit form the graphon features.

**Computational Complexity Analysis.** We now discuss computational complexity of the proposed $\mathcal{G}$-Mixup. The major additional computation costs come from graphon estimation and graph generation. *For graphon estimation*, suppose we have $M$ graphs and each of them has $N$ nodes and $E$ edges, and estimate step function with $K$ partitions to approximate a graphon, the complexity of used graphon estimation methods (Xu et al., 2021) is in Table 1. *For graph generation*, suppose we need to generate $K$ graphs with $N$ nodes, the computational complexity is $\mathcal{O}(KN)$ for node generation and $\mathcal{O}(KN^2)$ for edge generation, so the overall complexity of graph generation is $\mathcal{O}(KN^2)$.

Table 1: Computational complexity of graphon estimation (Xu et al., 2021)

| Method | Complexity |
|---|---|
| MC | $\mathcal{O}(N^3)$ |
| USVT | $\mathcal{O}(N^3)$ |
| LG | $\mathcal{O}(MN^2)$ |
| SBA | $\mathcal{O}(MKN \log N)$ |
| SAS | $\mathcal{O}(MN \log N + K^2 \log K^2)$ |

## 4 THEORETICAL JUSTIFICATION

In the following, we will theoretically prove that, *the synthetic graphs generated by G-Mixup will be a mixture of original graphs*. We first define the discriminative motif, and then we justify the graphon mixup operation (Equation 4) and graph generation operation (Equation 5) by analysing the homomorphism density of discriminative motifs of the original graphs and the synthetic graphs.

**Definition 1 (Discriminative Motif)** *A discriminative motif $F_G$ of graph $G$ is the subgraph, with the minimal number of nodes and edges, that can decide the class the graph $G$. Discriminative motifs $\mathcal{F}_{\mathcal{G}}$ is the set of the discriminative motif of every graph in the graph set $\mathcal{G}$.*

Intuitively, the discriminative motif is the key topology of a graph, by which the graph can be distinguished. We assume that *(i) every graph $G$ has a discriminative motif $F_G$*, and *(ii) a set of graphs $\mathcal{G}$ has a finite set of discriminative motifs $\mathcal{F}_{\mathcal{G}}$*. The goal of graph classification is to filter out structural noise in graphs (Fox & Rajamanickam, 2019) and recognize the key typologies (discriminative motifs) to predict the class label. For example, benzene (a compound in chemistry) is distinguished by the discriminative motif ⬡ (benzene ring).

### 4.1 WILL DISCRIMINATIVE MOTIFS $F_{\mathcal{G}}$ AND $F_{\mathcal{H}}$ EXIST IN $\lambda W_{\mathcal{G}} + (1 - \lambda)W_{\mathcal{H}}$ ?

We answer this question by exploring the difference in homomorphism density of discriminative motifs between the original graphon and mixed graphon. We propose the following theorem,

**Theorem 1** *Given two sets of graphs $\mathcal{G}$ and $\mathcal{H}$, the corresponding graphons are $W_{\mathcal{G}}$ and $W_{\mathcal{H}}$, and the corresponding discriminative motif set $\mathcal{F}_{\mathcal{G}}$ and $\mathcal{F}_{\mathcal{H}}$. For every discriminative motif $F_{\mathcal{G}} \in \mathcal{F}_{\mathcal{G}}$ and $F_{\mathcal{H}} \in \mathcal{F}_{\mathcal{H}}$, the difference between the homomorphism density of $F_{\mathcal{G}}/F_{\mathcal{H}}$ in the mixed graphon $W_{\mathcal{I}} = \lambda W_{\mathcal{G}} + (1 - \lambda)W_{\mathcal{H}}$ and that of the graphon $W_{\mathcal{H}}/W_{\mathcal{G}}$ is upper bounded by*

$$|t(F_{\mathcal{G}}, W_{\mathcal{I}}) - t(F_{\mathcal{G}}, W_{\mathcal{G}})| \leq (1 - \lambda)\mathrm{e}(F_{\mathcal{G}})||W_{\mathcal{H}} - W_{\mathcal{G}}||_{\square} ,$$
$$|t(F_{\mathcal{H}}, W_{\mathcal{I}}) - t(F_{\mathcal{H}}, W_{\mathcal{H}})| \leq \lambda \mathrm{e}(F_{\mathcal{H}})||W_{\mathcal{H}} - W_{\mathcal{G}}||_{\square} \qquad (8)$$

*where $\mathrm{e}(F)$ is the number of nodes in graph $F$ and $||W_{\mathcal{H}} - W_{\mathcal{G}}||_{\square}$ is the cut norm [2].*

---
[2] Details about cut norm are in Appendic A.1

*Proof Sketch.* The proof follows the derivation of Counting Lemma for Graphons (Lemma 10.23 in Lovász (2012)), which relates the homomorphism density and the cut distance $||W_{\mathcal{H}} - W_{\mathcal{G}}||$ of graphons. Specifically, we take the two graphons in this Lemma to deduce the bound of the difference of homomorphism densities of $W_{\mathcal{I}}$ and $W_{\mathcal{G}}/W_{\mathcal{H}}$. Detailed proof are in Appendix A.2. ∎

Theorem 1 suggests that the difference in the homomorphism densities of the mixed graphon and original graphons is upper bounded. Note that difference depends on the hyperparameter $\lambda$, the edge number $\mathrm{e}(F_{\mathcal{G}})/\mathrm{e}(F_{\mathcal{H}})$ and the cut norm $||W_{\mathcal{H}} - W_{\mathcal{G}}||_{\square}$. Since the $\mathrm{e}(F_{\mathcal{G}})/\mathrm{e}(F_{\mathcal{H}})$ and the cut norm $||W_{\mathcal{H}} - W_{\mathcal{G}}||_{\square}$ are decided by the dataset ( can be regraded as a constant), the difference in homomorphism densities will be decided by $\lambda$. On this basis, the corresponding label of the graphon is set to $\lambda \mathbf{y}_{\mathcal{G}} + (1-\lambda)\mathbf{y}_{\mathcal{H}}$. **Therefore, $\mathcal{G}$-Mixup can preserve the different discriminative motifs of the two different graphons into one mixed graphon.**

### 4.2 WILL THE GENERATED GRAPHS FROM GRAPHON $W_{\mathcal{I}}$ PRESERVE THE MIXTURE OF DISCRIMINATIVE MOTIFS?

Ideally, the generated graphs should inherit the homomorphism density of discriminative motifs from the graphon. To verify this, we propose the following theorem.

**Theorem 2** *Let $W_I$ be the mixed graphon , $n \geq 1$, $0 < \varepsilon < 1$, and let $F_I$ be the mixed discriminative motif, then the $W_{\mathcal{I}}$-random graph $\mathbb{G} = \mathbb{G}(n, W_{\mathcal{I}})$ satisfies*

$$\mathrm{P}\left(|t(F_{\mathcal{I}}, \mathbb{G}) - t(F_{\mathcal{I}}, W_{\mathcal{I}})| > \varepsilon\right) \leq 2\exp\left(-\frac{\varepsilon^2 n}{8\mathrm{v}(F_I)^2}\right). \tag{9}$$

Theorem 2 states that for any nonzero specified margin $\varepsilon$, no matter how small, with a sufficiently large samples of graph from mixed graphon, the homomorphism density of discriminative motif in synthetic graphs will approximately equal to that in graphon $t(F_{\mathcal{I}}, \mathbb{G}) \approx t(F_{\mathcal{I}}, W_{\mathcal{I}})$ with high probability. In other words, it reads the synthetic graphs will preserve the discriminative motif of the mixed graphon with a very high probability if the sample number $n$ is large enough. The detailed proof is presented in Appendix A.3. **Therefore, $\mathcal{G}$-Mixup can preserve the different discriminative motifs of the two different graphs into one mixed graph.**

### 4.3 DISCUSSION

This section discusses the differences and relations between $\mathcal{G}$-Mixup and other graph augmentation strategies, such as DropEdge (Rong et al., 2020), and Manifold Mixup (Wang et al., 2021).

**Relation to Edge Perturbation Method** . Edge perturbation methods is to randomly perturb the edges to improve the GNNs, inlcuding DropEdge (Rong et al., 2020), Graphon-based edge perturbation (Hu et al., 2021). DropEdge drops graph edges independently with a specified probability, aiming to prevent over-smoothing and over-fitting issues in GNNs. Graphon-based edge perturbation (Rong et al., 2020) improves the Dropedge by dropping edge based on an estimated probability. One of the limitations of such methods is that the edge permutation is based on the one individual graph, the graphs will not mix up. Edge perturbation methods will not mix the two discriminative motifs together and only randomly add remove some edges while the graphon will add some edges based on the graphon. DropEdge and Graphon-based edge perturbation (Hu et al., 2021) are special cases of $\mathcal{G}$-Mixup while setting different hyperparameter $\lambda$. *i) $\mathcal{G}$-Mixup will degenerate into Graphon-based edge perturbation*, while $\lambda = 0$ in Equation 4. The mathematical expression is $W_{\mathcal{I}} = W_{\mathcal{H}}, \{I_1, I_2, \cdots, I_k\} \overset{\text{i.i.d}}{\sim} \mathbb{G}(k, W_{\mathcal{I}}), \mathbf{y}_{\mathcal{I}} = \mathbf{y}_{\mathcal{H}}$. *ii) $\mathcal{G}$-Mixup will degenerate into DropEdge*, while setting $\lambda = 0$ and masking graphons with adjacency matrix $\mathbf{A}$ in Equation 4. The expression is $W_{\mathcal{I}} = \mathbf{A} \odot W_{\mathcal{H}}, \{I_1, I_2, \cdots, I_k\} \overset{\text{i.i.d}}{\sim} \mathbb{G}(k, W_{\mathcal{I}}), \mathbf{y}_{\mathcal{I}} = \mathbf{y}_{\mathcal{H}}$, where $\odot$ is element-wise multiplication.

**Relation to Manifold Mixup for Graph**. The methods proposed by Wang et al. (2021) is to adopt Manifold Mixup to graph data, which mix up graphs representation in the embedding space. The Manifold Mixup is to stabilize the model training by interpolating hidden representation. Interpolating hidden representation limits its applications by that 1) learning algorithms must have hidden representation of graphs and 2) models must be modified to adapt Manifold Mixup. In contrast, $\mathcal{G}$-Mixup is capable of generating synthetic graphs without modifying models. As a data augmentation method, our proposal has broader applications, e.g., creating graphs for graph contrastive learning.

## 5 EXPERIMENTS

We evaluate the performance of the proposed $\mathcal{G}$-Mixup in this section. First, we visualize graphons and graph generation results to investigate what $\mathcal{G}$-Mixup actually do on real-world datasets in Section 5.1 and Section 5.2. Then, we evaluate the effectiveness of $\mathcal{G}$-Mixup in graph classification with various datasets and GNN backbones in Section 5.3, as well as how it improve the robustness of GNNs against label corruption and adversarial examples in Section 5.4.

### 5.1 DOES DIFFERENT CLASSES OF REAL-WORLD GRAPHS HAVE DIFFERENT GRAPHONS?

We visualize the estimated graphon in Figure 2 to examine whether there are different graphons for different classes of graphs. **Observation 1: different classes of graphs have different graphons in real-world dataset.** As shown in Figure 2, the graphons of different class of graphs in one dataset are distinctly different. The graphons of IMDB-BINAERY in Figure 2 shows that the graphon of class 1 has larger dense area, which indicates that the graphs in this class have a more large communities than the graphs of class 0. The graphons of REDDIT-BINARY in Figure 2 shows that graphs of class 0 have one high-degree nodes while the graphs of class 1 have two. This observation validates that real-world graphs of different classes have distinctly different graphons, which lays a solid foundation for generating the mixture of graphs by mixing up graphons.

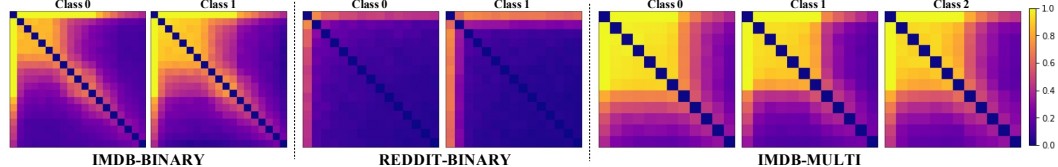

Figure 2: Estimated graphons on IMDB-BINARY, REDDIT-BINARY, and IMDB-MULTI. Obviously, graphons of different classes of graphs are quiet different. This observation validates the divergence of graphons between different classes of graphs, which is the basis of the $\mathcal{G}$-Mixup. The graphons are estimated by $LG$. More estimated graphons via various methods are in Appendix B.3.

### 5.2 WHAT IS $\mathcal{G}$-MIXUP DOING? A CASE STUDY

To investigate the outcome of $\mathcal{G}$-Mixup in real-world scenarios, we visualize the generated synthetic graphs in REDDIT-BINARY dataset in Figure 3. **Observation 2: synthetic graphs are indeed the mixture of the original graphs.** Original graphs and the generated synthetic graphs are visualized in Figure 3(a)(b) and Figure 3(c)(d)(e), respectively. Figure 3 demonstrates that mixed graphon $0.5 * W_{\mathcal{G}} + 0.5 * W_{\mathcal{H}}$ is able to generate graphs with a high-degree node and a dense subgraph, which can be regarded as the mixture of graphs with one high-degree node and two high-degree nodes. It validates that $\mathcal{G}$-Mixup prefer to preserve the discriminative motifs from the original graphs.

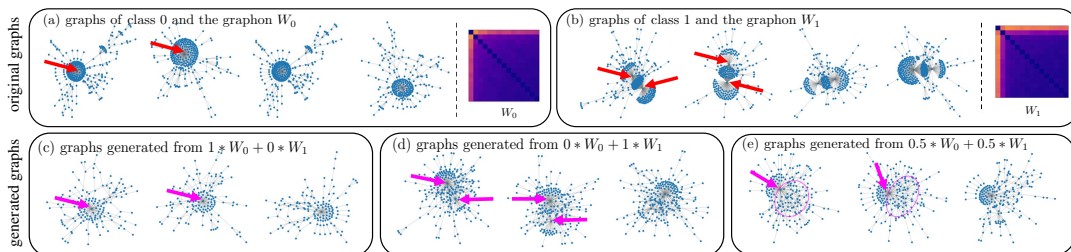

Figure 3: The visualization of generated synthetic graphs on dataset REDDIT-BINARY. The first row is the original graphs in the dataset while the second row is the generated graphs through the proposed $\mathcal{G}$-Mixup. The graphs in (a) and (b) are the original graphs of class 0 and class 1. The distinct difference of these two classes of graphs is that graphs of class 0 have one high-degree node while graphs of class 1 have two ( marked with → in (a) and (b) ). (c)/(d) shows graphs generated with the mixed graphon $(1 * W_0 + 0 * W_1)$ / $(0 * W_0 + 1 * W_1)$, which have one/two high-degree node/nodes (marked with → in (c) and (d)) because the mixed graphon only contains $W_0$/$W_1$. The synthetic graphs generated from $(0.5 * W_0 + 0.5 * W_1)$ is the mixture of graphs of class 0 and class 1, which appears as a high-degree node and a dense subgraph ( marked with → and ◯ in (e), respectively). The visualization shows that synthetic graphs are the mixture of the original graphs.

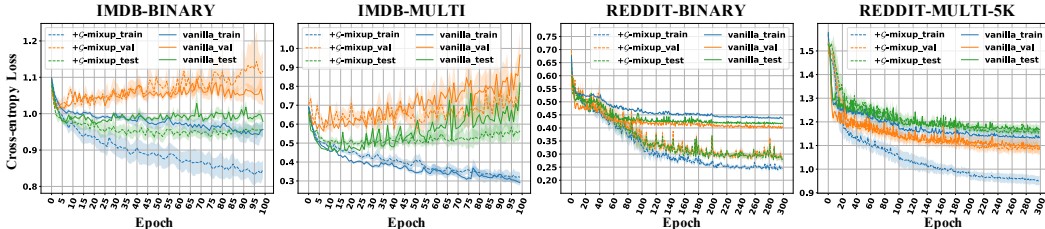

Figure 4: The training/validation/test curves on IMDB-BINARY, IMDB-MULTI, REDDIT-BINARY and REDDIT-MULTI-5K with GCN as backbone. The curves are depicted on ten runs.

Table 2: Performance comparisons of $\mathcal{G}$-Mixup with different graph neural networks on different dataset. The metric is the classification accuracy and its standard deviation. The best performance is in **boldface**. Experimental settings are in Appendix B.1. Experiments with more backbones (Diff-Pool, MincutPool, GMT) and molecular property prediction are in Appendix C.1 and Appendix C.2.

| | Dataset | IMDB-B | IMDB-M | REDDIT-B | REDD-M5k | REDD-M12k |
|---|---|---|---|---|---|---|
| | #graphs | 1000 | 1500 | 2000 | 4999 | 11929 |
| | #classes | 2 | 3 | 2 | 5 | 11 |
| | #avg.nodes | 19.77 | 13.00 | 429.63 | 508.52 | 391.41 |
| | #avg.edges | 96.53 | 65.94 | 497.75 | 594.87 | 456.89 |
| GCN | vanilla | $72.18_{\pm1.55}$ | $48.79_{\pm2.72}$ | $78.82_{\pm1.33}$ | $45.07_{\pm1.70}$ | $46.90_{\pm0.73}$ |
| | w/ Dropedge | $72.50_{\pm0.31}$ | $49.08_{\pm1.89}$ | $81.25_{\pm8.15}$ | $51.35_{\pm1.54}$ | $47.08_{\pm0.55}$ |
| | w/ NodeDropping | $72.00_{\pm4.09}$ | $48.58_{\pm2.85}$ | $79.25_{\pm0.35}$ | $49.35_{\pm1.80}$ | $47.93_{\pm0.64}$ |
| | w/ Subgraph | $68.50_{\pm4.76}$ | $49.58_{\pm2.61}$ | $74.33_{\pm2.88}$ | $48.70_{\pm1.63}$ | $47.49_{\pm0.93}$ |
| | w/ ManfoldMixup | $72.83_{\pm1.75}$ | $49.50_{\pm1.97}$ | $75.75_{\pm4.53}$ | $49.82_{\pm0.85}$ | $46.92_{\pm1.05}$ |
| | w/ $\mathcal{G}$-Mixup | $\mathbf{72.87}_{\pm3.85}$ | $\mathbf{51.30}_{\pm2.14}$ | $\mathbf{89.81}_{\pm0.74}$ | $\mathbf{51.51}_{\pm1.70}$ | $\mathbf{48.06}_{\pm0.53}$ |
| GIN | vanilla | $71.55_{\pm3.53}$ | $48.83_{\pm2.75}$ | $92.59_{\pm0.86}$ | $55.19_{\pm1.02}$ | $50.23_{\pm0.83}$ |
| | w/ Dropedge | $\mathbf{72.20}_{\pm1.82}$ | $48.83_{\pm3.02}$ | $92.00_{\pm1.13}$ | $55.10_{\pm0.44}$ | $49.77_{\pm0.76}$ |
| | w/ NodeDropping | $72.16_{\pm0.28}$ | $48.33_{\pm0.98}$ | $90.25_{\pm0.98}$ | $53.26_{\pm4.99}$ | $49.95_{\pm1.70}$ |
| | w/ Subgraph | $68.50_{\pm0.86}$ | $47.25_{\pm3.78}$ | $90.33_{\pm0.87}$ | $54.60_{\pm3.15}$ | $49.67_{\pm0.90}$ |
| | w/ ManfoldMixup | $70.83_{\pm1.04}$ | $49.88_{\pm1.34}$ | $90.75_{\pm1.78}$ | $54.95_{\pm0.86}$ | $49.81_{\pm0.80}$ |
| | w/ $\mathcal{G}$-Mixup | $71.94_{\pm3.00}$ | $\mathbf{50.46}_{\pm1.49}$ | $\mathbf{92.90}_{\pm0.87}$ | $\mathbf{55.49}_{\pm0.53}$ | $\mathbf{50.50}_{\pm0.41}$ |
| TopKPool | vanilla | $72.37_{\pm5.01}$ | $50.57_{\pm1.62}$ | $90.30_{\pm1.47}$ | $45.07_{\pm1.70}$ | $\mathbf{45.06}_{\pm1.70}$ |
| | w/ Dropedge | $71.75_{\pm2.18}$ | $48.75_{\pm2.94}$ | $88.96_{\pm1.90}$ | $\mathbf{47.43}_{\pm1.82}$ | $44.56_{\pm1.41}$ |
| | w/ NodeDropping | $69.16_{\pm1.04}$ | $48.50_{\pm2.50}$ | $81.33_{\pm4.48}$ | $46.15_{\pm2.28}$ | $44.49_{\pm1.15}$ |
| | w/ Subgraph | $67.83_{\pm4.01}$ | $50.83_{\pm2.38}$ | $86.08_{\pm2.12}$ | $45.75_{\pm2.47}$ | $46.18_{\pm1.53}$ |
| | w/ ManfoldMixup | $71.83_{\pm3.03}$ | $51.22_{\pm1.17}$ | $87.58_{\pm3.16}$ | $45.60_{\pm2.35}$ | $43.81_{\pm0.95}$ |
| | w/ $\mathcal{G}$-Mixup | $\mathbf{72.80}_{\pm3.33}$ | $\mathbf{51.30}_{\pm2.14}$ | $\mathbf{90.40}_{\pm0.89}$ | $46.48_{\pm1.70}$ | $43.72_{\pm1.65}$ |

## 5.3 CAN $\mathcal{G}$-MIXUP IMPROVE THE PERFORMANCE AND GENERALIZATION OF GNNS?

To validate the effectiveness of our proposal, we experiment to compare the performance with various backbones of GNNs on various datasets, and summarize results in Table 2 as well as the training curves in Figure 4. Our main observations are: **Observation 3: $\mathcal{G}$-Mixup can significantly improves the performance of various graph neural networks on various datasets.** From Table 2, our proposal gain 12 best performances among 15 reported accuracies, which substantially improve the performance of GNNs. Overall, our proposal performs 2.84% better than vanilla model. Note that $\mathcal{G}$-Mixup and baseline models adopt the same architecture of GNNs (e.g., layers, activation functions) and the same training hyperparameters (e.g., optimizer, learning rate). Considering both model performance and experimental setting, the improvement adequately validates the effectiveness of our proposal. **Observation 4: $\mathcal{G}$-Mixup can significantly improve the generalization of various backbones of graph neural networks.** From the loss curve on test data (green line) in Figure 4, the loss of test data of $\mathcal{G}$-Mixup (dashed green lines) are consistently lower than the vanilla model (solid green lines). Considering both the better performance and the better test loss curves, our proposal substantially is able to improve the generalization of GNNs. **Observation 5: $\mathcal{G}$-Mixup largely stabilizes the model training.** As shown in Table 2, $\mathcal{G}$-Mixup achieves 11 lower standard deviation among total 15 reported numbers than vanilla model. Additionally, the train/validation/test

curves of $\mathcal{G}$-Mixup (dashed line) in Figure 4 are more stable than vanilla model (solid line). These all indicate that our proposal $\mathcal{G}$-Mixup is capable of stabilizing the training of graph neural networks.

## 5.4 CAN $\mathcal{G}$-MIXUP VIRTUALLY IMPROVE THE ROBUSTNESS OF GNNS?

We investigate the two kinds of robustness of the proposed $\mathcal{G}$-Mixup, including *Label Corruption Robustness* and *Topology Corruption Robustness*, and report the results in Table 3 and 4, respectively. More experimental settings are presented in Appendix B.2. **Observation 6: $\mathcal{G}$-Mixup improves the robustness of graph neural networks.** Table 3 shows our proposal gains the better performance, indicating it is more robust to noisy label than vanilla baseline. Table 4 shows that $\mathcal{G}$-Mixup is more robust when graph topology is corrupted since the accuracy is consistently better than baselines. This can be an advantage of $\mathcal{G}$-Mixup when graph labels or topologies are noisy.

Table 3: Robustness to label corruption with different corruption ratio.

| Models | Methods | 10% | 20% | 30% | 40% |
|---|---|---|---|---|---|
| IMDB-B | vanilla | 72.30±3.67 | 69.43±4.80 | 63.65±8.87 | **55.21**±8.75 |
| | w/ Dropedge | 72.00±2.44 | 69.52±3.25 | 64.12±3.44 | 48.50±0.00 |
| | w/ ManfoldMixup | 71.87±3.56 | 69.03±4.85 | 65.62±9.89 | 48.50±0.00 |
| | w/ $\mathcal{G}$-Mixup | **72.56**±3.08 | **69.87**±5.41 | **65.50**±8.90 | 52.56±6.97 |
| REDDIT-B | vanilla | **73.90**±1.43 | 75.68±2.75 | 68.12±0.81 | 46.50±0.00 |
| | w/ Dropedge | 73.75±1.28 | 72.06±1.42 | 46.50±0.00 | 46.50±0.00 |
| | w/ ManfoldMixup | 71.96±1.97 | 76.00±2.24 | 54.43±1.09 | 46.50±0.00 |
| | w/ $\mathcal{G}$-Mixup | 71.94±3.00 | **76.34**±1.49 | **74.21**±1.85 | **53.50**±0.00 |

Table 4: Robustness to topology corruption with different corruption ratio.

| Models | Methods | 10% | 20% | 30% | 40% |
|---|---|---|---|---|---|
| Removing edges | vanilla | 77.96±3.71 | 67.59±5.73 | 64.96±8.87 | 65.71±8.31 |
| | w/ Dropedge | 74.40±2.26 | 65.12±3.51 | 65.93±2.32 | 57.87±4.14 |
| | w/ ManfoldMixup | 75.62±1.59 | 65.81±3.84 | 59.81±9.45 | 57.31±3.15 |
| | w/ $\mathcal{G}$-Mixup | **81.46**±3.08 | **71.12**±7.47 | **67.46**±8.90 | **66.25**±7.78 |
| Adding edges | vanilla | 76.12±5.73 | 74.37±6.48 | 72.31±2.69 | 72.00±2.92 |
| | w/ Dropedge | 70.53±1.47 | 70.18±1.29 | 71.18±1.53 | 70.90±1.53 |
| | w/ ManfoldMixup | 73.41±2.40 | 71.87±1.28 | 71.50±2.03 | 71.21±2.00 |
| | w/ $\mathcal{G}$-Mixup | **84.31**±3.21 | **82.21**±4.31 | **77.00**±2.25 | **75.56**±3.05 |

## 6 RELATED WORKS

**Graph Data Augmentation** Graph neural networks (GNNs) also achieve the state-of-the-art performance on graph classification task (Kipf & Welling, 2016; Veličković et al., 2017; Hamilton et al., 2017; Xu et al., 2018; Zhang et al., 2018). In parallel, graph data augmentation methods are also proposes to improve the performance of GNNs. There are three categories of graph data augmentation, including node perturbation (You et al., 2020), edge perturbation (Rong et al., 2020; You et al., 2020), and subgraph sampling (You et al., 2020). We have discussed the detailed differences to the mainstream graph augmentation method in Section 4.3. However, the common limitation of the existing graph data augmentation methods is that these methods are based on one single graph while $\mathcal{G}$-Mixup is able to augment new graphs using multiple input graphs. Besides, there are model-independent graph data augmentation methods (Zhou et al., 2020a; Zhao et al., 2021) and graph data augmentation method for node classification, which is not applicable while our methods is genral plug-in model-agnostic graph data augmentation methods for graph classification.

**Graphon Estimation.** Graphons and convergent graph sequences have been broadly studied in mathematics (Lovász, 2012; Lovász & Szegedy, 2006; Borgs et al., 2008) and have been applied to to network science (Avella-Medina et al., 2018; Vizuete et al., 2021) and graph neural networks (Ruiz et al., 2020a;b). There are tow lines of works to estimate step functions, one is based on stochastic block models, such as sorting-and-smoothing (SAS) method (Chan & Airoldi, 2014), stochastic block approximation (SBA) (Airoldi et al., 2013), "largest gap" (LG) (Channarond et al., 2012); another one is based on matrix decomposition, such as matrix completion (MC) (Keshavan et al., 2010), universal singular value thresholding (USVT) (Chatterjee et al., 2015). We estimate the step function using the above five methods (Xu et al., 2021) and the estimation are in Appendix B.3.

## 7 CONCLUSION

This work develops $\mathcal{G}$-Mixup to augment graph data. Unlike image data, graph data is irregular, unaligned and in non-Euclidean space, making it hard to be mixed up. However, the graphs within one class have the same generator (i.e., graphon), which is regular, well-aligned and in Euclidean space. Thus we turn to mix up the graphons of different classes then generate synthetic graphs. $\mathcal{G}$-Mixup is a new graph data augmentation algorithm which mix up the input graph to interpolate the topologies of different classes of graphs. A variety of experiments have shown that graph neural networks trained with $\mathcal{G}$-Mixup achieve better performance and generalization in terms of model accuracy and model loss, and improve the robustness to noisy labels and corrupted topologies.

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

# A    PROOF OF THEOREM

In the appendix, we first present the preliminaries in Appendix A.1. And then we present complete proof for Theorem 1 and Theorem 2 in Section A.3 and A.2, respectively.

## A.1    PRELIMINARIES

Cut norm (Lovász, 2012; Zhao, 2019) is used to measure structural similarity of two graphons. The definition of cut norm is as follow:

**Definition 2** *The cut norm of grapon $W$ is defined as*

$$\|W\|_\Box = \sup_{S,T \subset [0,1]} \left| \int_{S \times T} W(x,y) dx dy \right|, \tag{10}$$

where the supremum is taken over all measurable subsets S and T .

The following lemma follows the derivation of counting lemma for graphons, are known in the paper (Lovász, 2012). It will be used to prove the Theorem 1.

**Lemma 1** *Let $F$ be a simple graph and let $W, W' \in \mathcal{W}$. Then*

$$|t(F,W) - t(F,W')| \le e(F) \|W - W'\|_\Box \tag{11}$$

*Proof of Lemma 1.* The proof follows Zhao (2019). For an arbitrary simple graph $F$, by the triangle inequality we have

$$
\begin{aligned}
&|t(F,W) - t(F,W')| \\
&= \left| \int \left( \prod_{u_i v_i \in E} W(u_i, v_i) - \prod_{u_i v_i \in E} W'(u_i, v_i) \right) \prod_{v \in V} dv \right| \\
&\le \sum_{i=1}^{|E|} \left| \int \left( \prod_{j=1}^{i-1} W'(u_j, v_j) \left( W(u_i, v_i) - W'(u_i, v_i) \right) \prod_{k=i+1}^{|E|} W(u_k, v_k) \right) \prod_{v \in V} dv \right|
\end{aligned}
\tag{12}
$$

Here, each absolute value term in the sum is bounded by the cut norm $\|W - W'\|_\Box$ if we fix all other irrelavant variables (everything except $u_i$ and $v_i$ for the $i$-th term), altogether implying that

$$| t(F,W) - t(F,W')| \le e(F) \|W - W'\|_\Box \tag{13}$$

∎

**Lemma 2 (Corollary 10.4 in Lovász & Szegedy (2006))** *Let $W$ be a graphon, $n \ge 1$, $0 < \varepsilon < 1$, and let $F$ be a simple graph, then the $W$-random graph $\mathbb{G} = \mathbb{G}(n, W)$ satisfies*

$$P \left( |t(F, \mathbb{G}) - t(F, W)| > \varepsilon \right) \le 2 \exp \left( -\frac{\varepsilon^2 n}{8 v(F)^2} \right) \tag{14}$$

## A.2    PROOF OF THEOREM 1

We have the mixed graphon $W_\mathcal{I} = \lambda W_\mathcal{G} + (1-\lambda) W_\mathcal{H}$. Let $W = W_\mathcal{I}$, $W' = W_\mathcal{G}$, and $F = F_\mathcal{G}$ in Lemma 1, we have,

$$
\begin{aligned}
|t(F_\mathcal{G}, W_\mathcal{I}) - t(F_\mathcal{G}, W_\mathcal{G})| &\le e(F_\mathcal{G}) \|W_\mathcal{I} - W_\mathcal{G}\|_\Box \\
|t(F_\mathcal{G}, \lambda W_\mathcal{G} + (1-\lambda) W_\mathcal{H}) - t(F, W_\mathcal{G})| &\le e(F_\mathcal{G}) \|\lambda W_\mathcal{G} + (1-\lambda) W_\mathcal{H} - W_\mathcal{G}\|_\Box \\
&\le e(F_\mathcal{G}) \|(1-\lambda)(W_\mathcal{H} - W_\mathcal{G})\|_\Box
\end{aligned}
\tag{15}
$$

Recall that the cut norm $\|W\|_\Box = \sup_{S,T \subseteq [0,1]} \left| \int_{S \times T} W \right|$.

obviously, suppose $\alpha \in \mathbb{R}$, we have

$$\|\alpha W\|_\square = \sup_{S,T \subseteq [0,1]} \left| \int_{S \times T} \alpha W \right| = \sup_{S,T \subseteq [0,1]} \left| \alpha \int_{S \times T} W \right| = \alpha \|W\|_\square \tag{16}$$

Based on Equation 15 and Equation 16, we have

$$|t(F_\mathcal{G}, \lambda W_\mathcal{G} + (1-\lambda)W_\mathcal{H}) - t(F_\mathcal{G}, W_\mathcal{G})| \leq \mathrm{e}(F_\mathcal{G})\|(1-\lambda)(W_\mathcal{H} - W_\mathcal{G})\|_\square$$
$$\leq (1-\lambda)\mathrm{e}(F_\mathcal{G})\|W_\mathcal{H} - W_\mathcal{G}\|_\square \tag{17}$$

Similarly, let $W = W_\mathcal{I}$, $W' = W_\mathcal{H}$ and $F = F_\mathcal{H}$ in Lemma 1, We can also easily obtain

$$|t(F_\mathcal{H}, \lambda W_\mathcal{G} + (1-\lambda)W_\mathcal{H}) - t(F_\mathcal{H}, W_\mathcal{H})| \leq \lambda \mathrm{e}(F_\mathcal{H})\|W_\mathcal{H} - W_\mathcal{G}\|_\square \tag{18}$$

Equation 17 and 18 produce the upper bound in Theorem 8. ∎

### A.3 PROOF OF THEOREM 2

Let $F$ and $W$ be the discriminative motif $F_G$ and the mixed graphon $W_\mathcal{I}$ in Lemma 2, we will have

$$\mathrm{P}\left(|t(F_\mathcal{I}, \mathbb{G}) - t(F_\mathcal{I}, W_\mathcal{I})| > \varepsilon\right) \leq 2\exp\left(-\frac{\varepsilon^2 n}{8\mathrm{v}(F_\mathcal{I})^2}\right) \tag{19}$$

which produces the result in 9. ∎

### A.4 GRAPHONS ESTIMATION BY STEP FUNCTIONS

The proof follows Xu et al. (2021). A graphon can always be approximated by a step function in the cut norm (Frieze & Kannan, 1999).

Let $\mathcal{P} = (\mathcal{P}_1, .., \mathcal{P}_K)$ be a partition of $\Omega$ into $K$ measurable sets. We define a step function $W_\mathcal{P} : \Omega^2 \mapsto [0,1]$ as

$$W_\mathcal{P}(x,y) = \sum_{k,k'=1}^{K} w_{kk'} 1_{\mathcal{P}_k \times \mathcal{P}_{k'}}(x,y), \tag{20}$$

where each $w_{kk'} \in [0,1]$ and the indicator function $1_{\mathcal{P}_k \times \mathcal{P}_{k'}}(x,y)$ is 1 if $(x,y) \in \mathcal{P}_k \times \mathcal{P}_{k'}$, otherwise it is 0. The weak regularity lemma Lovász (2012) shown below guarantees that every graphon can be approximated well in the cut norm by step functions.

**Theorem 3 (Weak Regularity Lemma (Lemma 9.9 in (Lovász, 2012)) )** *For every graphon $W$ and $K \geq 1$, there always exists a step function $\mathbf{W}$ with $|\mathcal{P}| = K$ steps such that*

$$\|W - \mathbf{W}\|_\square \leq \frac{2}{\sqrt{\log K}} \|W\|_{L_2}. \tag{21}$$

## B EXPERIMENTS SETTING

### B.1 EXPERIMENTAL SETTING

To ensure a fair comparison, we use the same hyperparater for modeling training and the same architecture for vanilla model and other baselines. For model training, we use the Adam optimizer(Kingma & Ba, 2015). The initial learning rate is $0.01$ and will drop the learning rate by half every 100 epochs. The batch size is set to 128. We split the dataset into train/val/test data by $7 : 1 : 2$. The best epoch are determined by the best validation accuracy. Note that best test epoch is selected on a validation set. We also report the test accuracy on ten runs.

For architecture of graph neural networks, the details are listed as follows,

- **GCN** (Kipf & Welling, 2016). Four GNN layers and global mean pooling are applied. All the hidden units is set to 64. The activation is ReLU (Nair & Hinton, 2010).

- **TopKPool** (Gao & Ji, 2019). Three GNN layers and three TopK pooling are applied. A there-layer percetron are adopted to predict the labels. All the hidden units is set to 64. The activation is ReLU (Nair & Hinton, 2010).
- **GIN** (Xu et al., 2018). We apply five GNN layers and all MLPs have two layers. Batch normalization (Ioffe & Szegedy, 2015) is applied on every hidden layer. All hidden units are set to 64. The activation is ReLU (Nair & Hinton, 2010).

For hyperparemeter in $\mathcal{G}$-Mixup, we generate 20% more graph for training graph. The graphons are estimated based on the training graphs. We use different $\lambda \in [0.1, 0.2]$ to mix up the graphon and generate synthetic with different strength of mixing up.

## B.2 Experimental Setting of Robustness

The graph neural network adopted in this experiment is GCN, the architecture of which is as above. For label corruption, we randomly corrupt the graph labels with different corruption ratio $10\%, 20\%, 30\%, 40\%$. For topology corruption, we we randomly remove/add edges with different corruption ratio $10\%, 20\%, 30\%, 40\%$. The dataset for topology corruption is REDDIT-BINARY.

## B.3 Visualization of Graphons on More Real-World Dataset

$\mathcal{G}$-Mixup explores five graphon estimation methods, including sorting-and-smoothing (SAS) method (Chan & Airoldi, 2014), stochastic block approximation (SBA) (Airoldi et al., 2013), "largest gap" (LG) (Channarond et al., 2012), matrix completion (MC) (Keshavan et al., 2010) and the universal singular value thresholding (USVT) (Chatterjee et al., 2015). We present the estimated graphon by $LG$ in Figure 2. Here we present more visualization of graphons on IMDB-BINARY, REDDIT-BINARY and IMDB-MULTI dataset. An obvious observation is that graphons of different classes of graphs are different. This observation further validates the divergence of graphon between different classes of graphs.

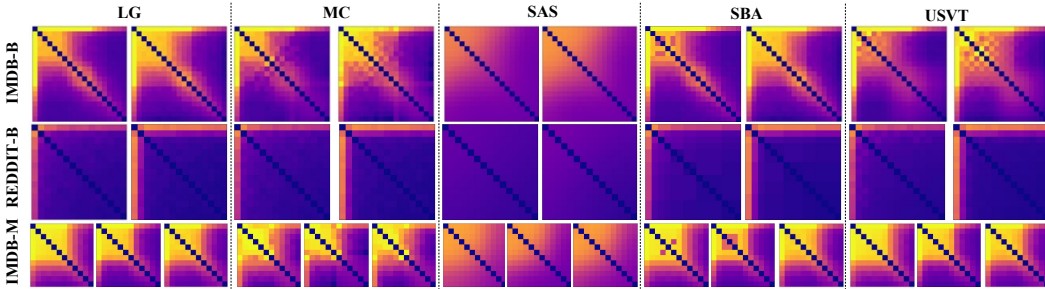

Figure 5: The estimated graphon on various dataset with different graphon estimation methods.

# C    ADDITIONAL EXPERIMENTS FOR REBUTTAL

In this appendix, we conduct additional experiments to further investigate the proposed method. The additional experiments include 1) more graph neural networks ((DiffPool, MincutPool, GMT)) in Appendix C.1, 2) molecular property prediction task with OGB datasets in Appendix C.2, 3) experiments on the impact of the nodes number of generated graphs in Appendix C.3 and 4) experiments on the performance of GCN with different layers in Appendix C.4.

## C.1    EXPERIMENT ON MORE GRAPH NEURAL NETWORKS (DIFFPOOL, MINCUTPOOL, GMT)

To further validate the effectiveness of $\mathcal{G}$-Mixup on more graph neural networks, we experiment with DiffPool (Ying et al., 2018), MincutPool (Bianchi et al., 2020) and GMT (Baek et al., 2020). For GMT, we use their released code and the recommended hyperparameters for their used datasets (D&D, MUTAG, PROTEINS, IMDB-B, IMDB-M) in their paper. To reproduce its results, we use their official code and the above datasets. The results are presented in Tables 5 and 6.

The details of backbones are listed as follows:

- **DiffPool** (Ying et al., 2018) is a differentiable graph pooling methods that can be adapted to various GNN architectures, which maps nodes to clusters based on their learned embeddings.
- **MincutPool** (Bianchi et al., 2020) is a differentiable pooling baselines. It learns a clustering function that can be quickly evaluated on out-of-sample graphs.
- **GMT** (Baek et al., 2020) is a multi-head attention based global pooling layer to generate graph representation, which captures the interaction between nodes according to their structure.

Our main observations are: **Observation 7: $\mathcal{G}$-Mixup improves the performance of DiffPool and MincutPool on various datasets.** From Table 5, our proposal gains 7 best performances among 8 reported accuracies, which substantially improve the performance of DiffPool and MincutPool. **Observation 8: $\mathcal{G}$-Mixup can significantly improve the performance of GMT.** Table 6 shows that $\mathcal{G}$-Mixup outperform all the baselines on all datasets. Overall, $\mathcal{G}$-Mixup outperform vanilla, Dropedge, ManifoldMixup by 1.44%, 1.28%, 2.01%, respectively. This indicates the superiority of $\mathcal{G}$-Mixup for graph classification task.

Table 5: Performance comparisons of $\mathcal{G}$-Mixup with DiffPool and MincutPool on different datasets. The metric is classification accuracy and its standard deviation. The best performance is in **boldface**.

| Backbone | Method | IMDB-B | IMDB-M | REDDIT-B | REDDIT-M5k |
|---|---|---|---|---|---|
| DiffPool | vanilla | $71.68_{\pm3.40}$ | $47.75_{\pm2.34}$ | $78.40_{\pm4.38}$ | $31.61_{\pm5.95}$ |
| | w/ Dropedge | $69.16_{\pm2.51}$ | $49.44_{\pm2.50}$ | $76.00_{\pm5.50}$ | $34.46_{\pm6.80}$ |
| | w/ NodeDropping | $70.25_{\pm3.01}$ | $46.83_{\pm1.34}$ | $76.68_{\pm2.57}$ | $33.10_{\pm5.53}$ |
| | w/ Subgraph | $69.50_{\pm2.16}$ | $46.00_{\pm4.43}$ | $76.06_{\pm2.81}$ | $31.65_{\pm4.43}$ |
| | w/ ManfoldMixup | $66.50_{\pm4.04}$ | $45.16_{\pm4.63}$ | $78.37_{\pm2.29}$ | $34.46_{\pm6.80}$ |
| | w/ $\mathcal{G}$-Mixup | $\mathbf{73.25}_{\pm3.89}$ | $\mathbf{50.70}_{\pm2.79}$ | $\mathbf{78.87}_{\pm2.27}$ | $\mathbf{38.42}_{\pm6.51}$ |
| MincutPool | vanilla | $73.25_{\pm3.27}$ | $49.04_{\pm3.57}$ | $84.95_{\pm3.25}$ | $49.32_{\pm2.67}$ |
| | w/ Dropedge | $69.16_{\pm2.51}$ | $49.66_{\pm1.73}$ | $81.37_{\pm1.59}$ | $47.20_{\pm1.10}$ |
| | w/ NodeDropping | $73.50_{\pm3.89}$ | $49.91_{\pm2.83}$ | $85.68_{\pm2.04}$ | $46.82_{\pm4.60}$ |
| | w/ Subgraph | $70.25_{\pm1.84}$ | $48.18_{\pm1.10}$ | $84.91_{\pm2.50}$ | $49.22_{\pm2.49}$ |
| | w/ ManfoldMixup | $70.62_{\pm2.09}$ | $49.96_{\pm1.86}$ | $85.12_{\pm2.29}$ | $47.20_{\pm1.10}$ |
| | w/ $\mathcal{G}$-Mixup | $\mathbf{73.93}_{\pm2.84}$ | $\mathbf{50.29}_{\pm2.30}$ | $\mathbf{85.87}_{\pm1.37}$ | $\mathbf{50.12}_{\pm2.47}$ |

Table 6: Performance comparisons of $\mathcal{G}$-Mixup with GMT on different dataset. The metric is the classification accuracy and its standard deviation. The best performance is in **boldface**.

| Backbone | Method | D&D | MUTAG | PROTEINS | IMDB-B | IMDB-M |
|---|---|---|---|---|---|---|
| GMT | vanilla | $78.29_{\pm5.77}$ | $82.77_{\pm6.30}$ | $74.59_{\pm5.29}$ | $73.60_{\pm3.87}$ | $50.73_{\pm3.03}$ |
| | w/ Dropedge | $78.37_{\pm4.17}$ | $82.22_{\pm8.88}$ | $74.32_{\pm5.42}$ | $73.40_{\pm3.85}$ | $50.73_{\pm3.09}$ |
| | w/ ManfoldMixup | $77.69_{\pm3.81}$ | $82.22_{\pm10.48}$ | $74.41_{\pm3.97}$ | $73.70_{\pm3.79}$ | $49.93_{\pm3.49}$ |
| | w/ $\mathcal{G}$-Mixup | $\mathbf{79.57}_{\pm3.69}$ | $\mathbf{84.44}_{\pm8.88}$ | $\mathbf{75.13}_{\pm5.06}$ | $\mathbf{74.70}_{\pm3.76}$ | $\mathbf{51.33}_{\pm3.52}$ |

## C.2 EXPERIMENT ON MOLECULAR PROPERTY PREDICTION

We experiment on molecular property prediction task (Hu et al., 2020), including ogbg-molhiv, ogbg-molbace, ogbg-molbbbp. In these dataset, each graph represents a molecule, where nodes are atoms, and edges are chemical bonds. We adopte official reference graph neural network backbones (gcn, gcn-vitual, gin, gin-vitual) as our backbones, and we generate the edge attributes randomly for synthetic graphs. The results are presented in Table 7. **Observation 9: $\mathcal{G}$-Mixup can improve the performance of GNNs on molecular property prediction task with the experimental setting for a fair comparison.** Table 7 shows that $\mathcal{G}$-Mixup gains 9 best performances among 12 reported AUCs.

Table 7: Performance comparisons of $\mathcal{G}$-Mixup on molecular property prediction task. The metric is AUROC [3] and its standard deviation. The best performance is in **boldface**.

| Backbones | Mehtods | ogbg-molhiv | ogbg-molbbbp | ogbg-molbace |
|---|---|---|---|---|
| GCN | vanilla | $76.24_{\pm 0.98}$ | $68.05_{\pm 1.52}$ | $80.36_{\pm 1.56}$ |
| | w/ Dropedge | $75.93_{\pm 0.76}$ | $68.02_{\pm 0.95}$ | $80.22_{\pm 1.59}$ |
| | w/ ManifoldMixup | $76.24_{\pm 1.40}$ | $68.36_{\pm 2.05}$ | $80.46_{\pm 2.05}$ |
| | w/ G-Mixup | $\mathbf{76.29}_{\pm 0.80}$ | $\mathbf{68.45}_{\pm 0.84}$ | $\mathbf{80.73}_{\pm 2.06}$ |
| GCN-virtual | vanilla | $75.62_{\pm 1.65}$ | $65.13_{\pm 1.11}$ | $\mathbf{74.49}_{\pm 3.04}$ |
| | w/ Dropedge | $74.64_{\pm 1.32}$ | $66.46_{\pm 1.61}$ | $69.75_{\pm 3.47}$ |
| | w/ ManifoldMixup | $74.04_{\pm 2.06}$ | $65.51_{\pm 1.74}$ | $73.10_{\pm 4.97}$ |
| | w/ G-Mixup | $\mathbf{76.56}_{\pm 0.80}$ | $\mathbf{67.20}_{\pm 1.30}$ | $73.55_{\pm 4.79}$ |
| GIN | vanilla | $77.08_{\pm 1.96}$ | $68.42_{\pm 2.31}$ | $75.91_{\pm 1.01}$ |
| | w/ Dropedge | $75.77_{\pm 1.75}$ | $66.16_{\pm 2.96}$ | $70.50_{\pm 6.24}$ |
| | w/ ManifoldMixup | $75.73_{\pm 1.25}$ | $68.15_{\pm 2.04}$ | $77.44_{\pm 4.13}$ |
| | w/ G-Mixup | $\mathbf{77.14}_{\pm 0.45}$ | $\mathbf{69.28}_{\pm 1.24}$ | $\mathbf{77.79}_{\pm 3.34}$ |
| GIN-virtual | vanilla | $\mathbf{77.52}_{\pm 1.56}$ | $67.10_{\pm 2.10}$ | $\mathbf{74.19}_{\pm 4.99}$ |
| | w/ Dropedge | $76.83_{\pm 1.11}$ | $68.87_{\pm 1.17}$ | $72.20_{\pm 3.37}$ |
| | w/ ManifoldMixup | $76.51_{\pm 2.22}$ | $68.04_{\pm 2.87}$ | $74.17_{\pm 1.38}$ |
| | w/ G-Mixup | $77.09_{\pm 1.07}$ | $\mathbf{70.02}_{\pm 1.68}$ | $73.53_{\pm 3.98}$ |

## C.3 WHAT IS THE IMPACT OF NUMBER OF NODES OF GENERATED GRAPHS?

We investigate the impact of number of nodes of generated synthetic graphs by $\mathcal{G}$-Mixup and report the results in Figure 6. Specifically, $\mathcal{G}$-Mixup generates synthetic graphs with different the numbers (hyperparameters $K$) of nodes and use them to train graph neural networks. From Figure 6, we can see that average number of nodes in the original graphs is a better choice for hyperparameters $K$ for $\mathcal{G}$-Mixup, which is accords with intuition.

## C.4 HOW $\mathcal{G}$-MIXUP PERFORM WHEN GRAPH NEURAL NETWORK GOES DEEPER?

We investigate the performance of $\mathcal{G}$-Mixup when the GCN goes deeper and report the results in Figure 7. We experimented with different numbers $(2-9)$ of layers to investigate the performance of $\mathcal{G}$-Mixup. **Observation 10: $\mathcal{G}$-Mixup improves the performance of graph neural networks with different layers**. The left figure in Figure 7 shows $\mathcal{G}$-Mixup gains better performance for IMDB-BINARY dataset while the depth of GCNs is $2-6$. The performance with deeper GCN $(7-9)$ are comparable to baselines, however, the accuracy of deeper GCN is much lower than shallow ones. The right figure in Figure 7 shows $\mathcal{G}$-Mixup gains better performance by a significant margin for REDDIT-BINARY dataset while the depth of GCNs is $2-9$. This validates the effectiveness of $\mathcal{G}$-Mixup when graph neural network goes deeper.

---

[3]Area Under Receiver Operating Characteristic

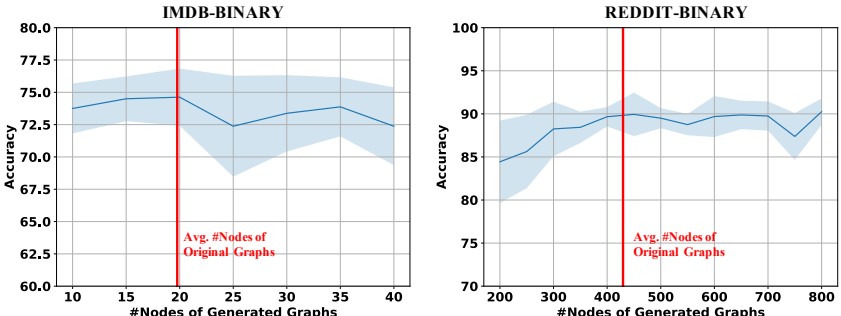

Figure 6: The impact of the node numbers of generated synthetic graphs on IMDB-BINARY and REDDIT-BINARY datasets. The red vertical line indicates the average number of all the original graphs. The blue line represents that classification accuracy with different number of nodes of generated graphs. Obviously, the accuracy reach the maximum values around the red line on both two datasets

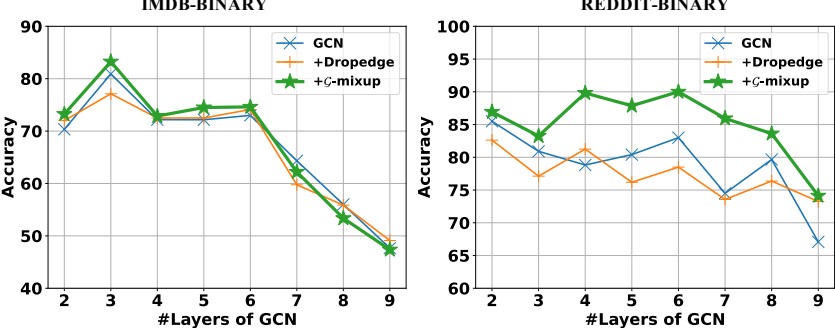

Figure 7: The performance of $\mathcal{G}$-mixup using GCNs with different depth on IMDB-BINARY and REDDIT-BINARY. This figure show that $\mathcal{G}$-Mixup consistently improve GCN when it goes deeper.

