# OpenReview forum: "G-Mixup: Graph Augmentation for Graph Classification"
_ICLR.cc/2022/Conference — ICLR 2022 Submitted_

### Official Review · Reviewer_vVum · 2021-11-01

**Correctness:** 3
**Technical Novelty And Significance:** 3
**Empirical Novelty And Significance:** 3
**Recommendation:** 8
**Confidence:** 4

**Main Review:**

## Strengths

* Clear contribution that generalizes the mixup [1] to the graph data. Although mixup has shown to be an effective data augmentation method in the image and text domain, it has not been applied to the graph domain due to the non-Euclidean characteristic of a graph. Therefore, the contribution of this work is clear since this work makes the mixup be applicable to the graph-structured data by introducing the concept of the graphon.
* Well-written. Especially, it is easy to understand the concept of this paper through the figures and examples. The theoretical justification of the method clearly shows the G-mixup is effective to generate the mixup of two graphs of different classes on the theoretical side. In addition, the analysis in Section 5.2 empirically shows that the G-mixup can generate a well-mixed graph of different classes.

## Weaknesses
The empirical results are not sufficient to show the effectiveness of G-mixup.
* Baselines are not enough. I wonder that why the authors did not include the experiments against Manifold Mixup [2] although the authors refer to it in the discussion.
* Besides, several other baselines [3,4,5] including simple augmentations should be compared against the G-mixup.
* Experimental setting is too limited. Since the G-mixup mainly focuses on the improvement of graph neural networks on the graph classification task, it is important to show its effectiveness in various settings including dataset and base models. For instance, I recommend authors include the experiments on the classification task of a biomedical graph (e.g. molhiv dataset in Open Graph Benchmark [6]).
* In terms of the base GNN model, I believe the experiments on the GNN with the recent pooling [7,8] method will also be helpful to show the effectiveness of G-mixup for the performance improvement on the classification task.

Overall, the experiment part should be improved.



### References
[1] Zhang et al., mixup: Beyond Empirical Risk Minimization, ICLR 2018.

[2] Wang et al., Mixup for Node and Graph Classification, WWW 2021.

[3] Zhao et al., Data Augmentation for Graph Neural Networks, AAAI 2021.

[4] You et al., Graph Contrastive Learning with Augmentations, NeurIPS 2020.

[5] Zhou et al., Data Augmentation for Graph Classification, CIKM 2020.

[6] Hu et al., Open Graph Benchmark: Datasets for Machine Learning on Graphs, NeurIPS 2020.

[7] Bianchi et al., Spectral Clustering with Graph Neural Networks, ICML 2020.

[8] Baek et al., Accurate Learning of Graph Representations with Graph Multiset Pooling, ICLR 2021.


**Summary Of The Paper:**

This paper proposes the mixup method for graphs which is based on the concept of the graphon. Specifically, G-mixup first estimates the graphon for each class of graphs with the same label then interpolates the graphons of different classes of graphs. Finally, they generate the new graphs from the interpolated graphons to produce the mixed graph with a mixed label.

The authors theoretically show that G-mixup can preserve the different motifs of the two different graphs into one mixed graphon. In experiments, they empirically show that the G-mixup improve the generalization and the performance of existing GNNs on several graph classification task.


**Summary Of The Review:**

This is an impressive paper but there is a lack of experiments.
I am willing to upraise my score if the authors successfully address my concerns regarding experiments in the rebuttal.

------------
After rebuttal: In this version, the experimental results seem convincing. I raise my score from 5 to 8.

---

> ### Author Response · Authors · 2021-11-17
> **Response to Reviewer vVum ( part 2 / 2 )**
>
> > **3. ...I recommend authors include the experiments on the classification task of a biomedical graph (e.g. molhiv dataset in Open Graph Benchmark).**
>
> - We added the molecular property prediction task, including **ogbg-molhiv**, ogbg-molbbbp, and ogbg-molbace datasets. The additional experiments are presented in **Appendix C.2**. Since the OGB dataset has official reference graph neural network backbones (gcn, gcn-vitual, gin, gin-vitual)[11], we adopt them as graph neural networks. And we also adopted the vanilla model, DropEdge, and ManifoldMixup as our baselines. The evaluation metric is AUROC. The results show **G-Mixup can improve the performance of GNNs on molecular property prediction task.**
>
> | Backbones  | Mehtods | ogbg-molhiv |ogbg-molbbbp|ogbg-molbace|
> | :--- |   :---- | :---: | :---:|  :---:  |
> |   gcn          | vanilla              | 76.24±0.98  | 68.05±1.52  | 80.36±1.56  |
> |                | w/ Dropedge          | 75.93±0.76  | 68.02±0.95  | 80.22±1.59  |
> |                | w/ ManifoldMixup     | 76.24±1.40  | 68.36±2.05  | 80.46±2.05  |
> |                | w/ G-Mixup           | **76.29**±0.80  | **68.45**±0.84  | **80.73**±2.06  |
> |   gcn-virtual  | vanilla              | 75.62±1.65  | 65.13±1.11  | 74.49±3.04  |
> |                | w/ Dropedge          | 74.64±1.32  | 66.46±1.61  | 69.75±3.47  |
> |                | w/ ManifoldMixup     | 74.04±2.06  | 65.51±1.74  | 73.10±4.97  |
> |                | w/ G-Mixup           | **76.56**±0.80  | **67.20**±1.30  | **73.55**±4.79  |
> |   gin          | vanilla              | 77.08±1.96  | 68.42±2.31  | 75.91±1.01  |
> |                | w/ Dropedge          | 75.77±1.75  | 66.16±2.96  | 70.50±6.24  |
> |                | w/ ManifoldMixup     | 75.73±1.25  | 68.15±2.04  | 77.44±4.13  |
> |                | w/ G-Mixup           | **77.14**±0.45  | **69.28**±1.24  | **77.79**±3.34  |
> |   gin-virtual  | vanilla              | **77.52**±1.56  | 67.10±2.10  | **74.19**±4.99  |
> |                | w/ Dropedge          | 76.83±1.11  | 68.87±1.17  | 72.20±3.37  |
> |                | w/ ManifoldMixup     | 76.51±2.22  | 68.04±2.87  | 74.17±1.38  |
> |                | w/ G-Mixup           | 77.09±1.07  | **70.02**±1.68  | 73.53±3.98  |
>
>
>
> > **4.Other additional experiments**
>
> - To further investigate the properties of G-mixup, we add more experiments as follows:
>   - We added the experiments on the impact of the number of nodes of the generated synthetic graphs by generating graphs with different nodes. The result shows that the **average number of nodes in all original graphs is a better choice for generating nodes**. The experiments are presented in  **Appendix C.3**.
>   - We added the performance of G-mixup when graph neural networks go deeper. The results show **G-Mixup improves the performance of graph neural networks with different layers**. The experiments are presented in **Appendix C.4**.
>   - We added more baselines (Dropedge and ManifoldMixup) for comparison of robustness. The results show that **G-Mixup is more robust than DropEdge and ManifoldMixup** in the context of label corruption and topology corruption. The experiments are presented in **Section 5.4**.
>
> Thank you again for your constructive comments. We are looking forward to your feedback and are glad to answer your follow-up questions.
>
>
> **[References]**
>
> [1] You et al., Graph Contrastive Learning with Augmentations, NeurIPS 2020.
>
> [2] Wang et al., Mixup for Node and Graph Classification, WWW 2021.
>
> [3] Hu et al., Open Graph Benchmark: Datasets for Machine Learning on Graphs, NeurIPS 2020.
>
> [4] Ying et al., Hierarchical graph representation learning with differentiable pooling. NeurIPS 2018.
>
> [5] Bianchi et al., Spectral Clustering with Graph Neural Networks, ICML 2020.
>
> [6] Baek et al., Accurate Learning of Graph Representations with Graph Multiset Pooling, ICLR 2021.
>
> [7] https://github.com/vanoracai/MixupForGraph
>
> [8] https://github.com/pyg-team/pytorch_geometric/blob/1.7.2/examples/proteins_diff_pool.py
>
> [9] https://github.com/pyg-team/pytorch_geometric/blob/1.7.2/examples/proteins_mincut_pool.py
>
> [10] https://github.com/JinheonBaek/GMT
>
> [11] https://github.com/snap-stanford/ogb/tree/master/examples/graphproppred/mol

---

> > ### Comment · Reviewer_vVum · 2021-11-21
> > **Thank you for the response and sorry for the late response**
> >
> > I sincerely appreciate the detailed response of the authors for my concerns and extensive experiments that the authors try to alleviate the concerns on the lack of experimental results.
> > I tried to thoroughly read your responses and the updated paper.
> >
> > Here are my comments on the rebuttal:
> > ### Pros
> > * Compared to the very first submission, additional experimental results make the proposed method, G-mixup, more convincible by showing the performance improvement on various datasets including the molecular property prediction tasks, which are not addressed before. Results also show that G-mixup can generalize to various types of graph data.
> > * The authors add three baselines which are simple but effective augmentation methods for the graph data, which were missing in the original version. The results against such methods show that the G-mixup can be a powerful and practical data augmentation method for graph data.
> > * Experiments with the modern pooling method also show that G-mixup fundamentally improves the performance of the Graph Neural Network on graph classification task, by successfully augmenting the data.
> >
> > ### Cons
> > * In some experimental results, it is suspect whether the improvements from G-mixup are statistically significant (e.g., Table 2 IMDB-B, Table 7 ogbg-molhiv). It will be appreciated if the authors include the statistical significance test on each experiment in the revision.
> >
> > Overall, I am surely satisfied with the additional experiments and their results during the rebuttal phase. Although the improvements of G-mixup seem marginal in some experiments, it has shown to be effective in general compared to previous data augmentation methods on various datasets and backbone models.
> >
> > I believe that G-mixup has enough novelty and practicability to be accepted and will contribute to the community by suggesting a promising line of research on the data augmentation of the graph data. Therefore, I upraise my score from 5 to 8.

---

> > > ### Author Response · Authors · 2021-11-22
> > > **Thanks very much for your positive feedback**
> > >
> > > Dear Reviewer vVum,
> > >
> > > Thanks for your positive re-assessment of our work.  We enjoyed the fruitful discussion with you, and we will revise our work carefully based on your suggestions. Please do let us know if there is anything that you believe we can do to improve it!
> > >
> > > Sincerely, Authors

---

> ### Author Response · Authors · 2021-11-17
> **Response to Reviewer vVum ( part 1 / 2 )**
>
> Thanks for your valuable suggestions and for appreciating the novelty and contributions of our work. We believe your suggestions on our experiments would significantly improve our work. To address your concerns about the experiments, we improve the experiments by adding
> - more baselines (NodeDropping[1], SubGraph[1], ManifoldMixup[2]),
> - molecular property prediction task (ogbg-molhiv, ogbg-molbbbp,ogbg-molbace)[3],
> - more graph neural networks (DiffPool[4], MincutPool[5], GMT[6]),
> - more baselines (Dropedge and ManifoldMixup) for comparison of robustness.
> - experiments on the impact of the nodes number of the generated graphs,
> - experiments on the performance of GCN with different layers.
>
> Below we address your concerns detailedly.
>
> > **1. I wonder that why the authors did not include the experiments against Manifold Mixup. Besides, several other baselines including simple augmentations.**..
>
> - We added Manifold Mixup[2]，Nodedropping[1] and Subgraph[1] as baselines.  Thus our baselines include the mainstream graph data augmentation methods(node, edge, and subgraph perturbation). Among all the baselines, **G-Mixup achieves the best performance on the graph classification task.** Please check the updated **Tables 2 and 5**.
> - We did not include Manifold Mixup in the previous version because the public source code [7] of Manifold Mixup lacks code for graph classification task. To address your concern, we implemented it by ourselves and reported its performance this time. In our implementation, we interpolate the last latent representations before the output layer and then interpolate the one-hot labels. We also keep the same augmentation ratio as $0.15$ to G-mixup and use the identical backbones and training process as other baselines to ensure a fair comparison.
>
>
> > **2. ..experiments on the GNN with the recent pooling method will also be helpful to show the effectiveness of G-mixup..**
>
> - We added modern pooling methods, including DiffPool[4], MincutPool[5], and GMT[6], as our backbones. The additional experiments are presented in **Appendix C.1**.
> - For DiffPool and MincutPool, the adopted code for DiffPool backbone is at [8] and the code for MincutPool at [9]. The results show **G-Mixup improves the performance of DiffPool and MincutPool** on various datasets since it gains $7$ best performances among $8$ reported accuracies.
>
> - For GMT, we use their released code [10] and the recommended hyperparameters for used datasets (D&D, MUTAG, PROTEINS, IMDB-B, IMDB-M) in their paper. To reproduce its results, we use their official code and the above datasets. The results show **G-Mixup can significantly improve the performance of GMT** and it outperforms vanilla, Dropedge, ManifoldMixup by $1.44\%$, $1.28\%$, $2.01\%$, respectively.
>
> The results are as follows, and please check Appendix C.1 for more details.
>
> | Backbones  | Methods | IMDB-B|IMDB-M|REDDIT-B|REDDIT-M5k|
> | :--- |   :---- | :---: | :---:|  :---:  | :---:  |
> | DiffPool  | vanilla              | 71.68±3.40 | 47.75±2.34 | 78.40±4.38 | 31.61±5.95  |
> |           | w/ Dropedge          | 69.16±2.51 | 49.44±2.50 | 76.00±5.50 | 34.46±6.80  |
> |           | w/ NodeDropping      | 70.25±3.01 | 46.83±1.34 | 76.68±2.57 | 33.10±5.53  |
> |           | w/ Subgraph          | 69.50±2.16 | 46.00±4.43 | 76.06±2.81 | 31.65±4.43  |
> |           | w/ ManifoldMixup     | 66.50±4.04 | 45.16±4.63 | 78.37±2.29 | 34.46±6.80  |
> |           | w/ G-Mixup           | **73.25**±3.89 | **50.70**±2.79 | **78.87**±2.27 | **38.42**±6.51  |
> | MinCutPool| vanilla              | 73.25±3.27 | 49.04±3.57 | 84.95±3.25 | 49.32±2.67  |
> |           | w/ Dropedge          | 69.16±2.51 | 49.66±1.73 | 81.37±1.59 | 47.20±1.10  |
> |           | w/ NodeDropping      | 73.50±3.89 | 49.91±2.83 | 85.68±2.04 | 46.82±4.60  |
> |           | w/ Subgraph          | 70.25±1.84 | 48.18±1.10 | 84.91±2.50 | 49.22±2.49  |
> |           | w/ ManifoldMixup     | 70.62±2.09 | 49.96±1.86 | 85.12±2.29 | 47.20±1.10  |
> |           | w/ G-Mixup           | **73.93**±2.84 | **50.29**±2.30 | **85.87**±1.37 | **50.12**±2.47  |
>
>
>
> | Backbones  | Methods | DD |MUTAG|PROTEINS|IMDB-B|IMDB-M|
> | :--- |   :---- | :---: | :---:|  :---:  | :---:  | :---:  |
> |  GMT      | vanilla              | 78.29±5.77 | 82.77±6.30 | 74.59±5.29 | 73.60±3.87 | 50.73±3.03 |
> |           | w/ Dropedge          | 78.37±4.17 | 82.22±8.88 | 74.32±5.42 | **74.90**±4.45 | 50.73±3.09 |
> |           | w/ ManifoldMixup     | 77.69±3.81 | 82.22±10.48| 74.41±3.97 | 73.70±3.79 | 49.93±3.49 |
> |           | w/ G-Mixup           | **79.57**±3.69 | **84.44**±8.88 | **75.13**±5.06 | 74.70±3.76 | **51.33**±3.52 |

---

> ### Author Response · Authors · 2021-11-18
> **Looking forward to your feedback**
>
> Dear  Reviewer vVum,
>
> Thank you for your valuable suggestions again. We have responded to your initial comments. We are looking forward to your feedback and are glad to answer your further questions.
>
> Thanks, Authors

---

> ### Author Response · Authors · 2021-11-19
> **Looking forward to your feedback**
>
> Dear Reviewer vVum,
>
> Thank you for your valuable suggestions again. We have responded to your initial comments. We are looking forward to your feedback and are glad to answer your further questions.
>
> Thanks, Authors

---

> ### Author Response · Authors · 2021-11-21
> **Looking forward to your feedback**
>
> Dear Reviewer vVum,
>
> Thank you for your valuable suggestions again. We have responded to your initial comments. We are looking forward to your feedback and are glad to answer your further questions.
>
> Thanks, Authors

---

### Official Review · Reviewer_RkbU · 2021-11-01

**Correctness:** 3
**Technical Novelty And Significance:** 3
**Empirical Novelty And Significance:** 2
**Recommendation:** 3
**Confidence:** 4

**Main Review:**

The paper introduces a novel attempt of applying Mixup from vision and text to graph data. The idea of leveraging graphs being sampled from estimated graphons sounds interesting to me. On the other hand, I have the following concerns regarding the paper.

1.	The proposed method bases on a strong assumption which deserves further justification. That is, each class has a graphon and this graphon can be accurately estimated. I would like to see some theoretical proof and/or analysis on that. Such assumption also raises the question whether such graphon estimation can be used for graphs with node and edge features; it would be useful if the authors could shed some lights on that.

2.	Theorem 2 suggests that in order to allow the synthetic graphs sampled from the mixed graphons to preserve the discriminative motifs, one needs sufficiently large sampled graphs. This assumption of g-Mixup makes me wonder that at the earlier stage of training, the synthetic graphs cannot preserve the discriminative motif of the graphon. I wonder how this guarantees the proposed g-Mixup would work. In other words, the synthetic graphs may be noise at all. Also, since the addition and deletion of graph edge in g-Mixup are based on the learned (imperfect) graphon, I think it may encounter the manifold intrusion issue when mixing graphs (Guo et al. at “mixup as locally linear out-of-manifold regularization”, AAAI2019).

3.	The experiment section is quite weak in its current form.

     a.	A critical baseline is missed in the experiment. The proposed g-Mixup is closely related to the MixupGraph approach as proposed by Wang et al. (“Mixup for Node and Graph Classification”, WWW2021). Mixing the embeddings resulting from the READOUT function of GNNs is much easier to be implemented than the g-Mixup method proposed here, so it would be interesting to see the comparison between the two approaches.

     b.	It would be useful to provide ablation studies to show how accurate are the graphons estimated for the datasets (namely Equation 3), and how such estimation impacts the performance of the g-Mixup.

     c.	There is a parameter n, i.e., the number of nodes to be generate by the graphon. How this number impacts the performance of g-Mixup?

     d.	When estimating graphons from training data, K is used as the average number of nodes in all graphs. How the K was set in testing is not clear to me, and how different Ks impact the g-Mixup performance?

     e.	The datasets used are without nature node features, I wonder how the proposed method would perform on graphs with node features.

      f.	From Table2. The accuracy improvement over vanilla baselines is minor, in particular for the powerful model GIN. This makes me wonder why one needs extra computation cost for g-Mixup?

     g.	Also, for dropEdge, 4 layer GCN and 5 Layer GIN seem a bit too shallow to me. DropEdge typically performs well on deeper GNNs such as six to eight layers.

     h.	In Table3, it would be useful to also compare with dropEdge and GraphMixup.


Other comments:
1.	The sentence at the end of page one: “thus prohibit us from directly adopting the Mixup strategy to graph data”. “Prohibit” here is a bit strong to me, although I think it is quite challenge to applying Mixup to graph input.
2.	Last paragraph in section 1. “Since directly mixing up graphs is intractable,”. It would be helpful to make it specific why and how the intractable occurs here?
3.	Notations are inconsistent. Such as N(v_{i}) vs. N(i); k and n refer to nodes and graphs in different places (see end of section4.2).
4.	Typos: first sentence of section 4.2 “propose the following the theorem.”; middle of second paragraph in section 4.3: “on a estimated…”.


**Summary Of The Paper:**

The paper proposes to leverage graphon as a graph generator to adapt Mixup from vision and text to graph classification. It works as follows. A graphon, which aims to capture the existence probability of an edge between any two nodes in a graph, is first estimated for each class of the training graphs. Next, two random graphons are mixed with mixing ratio \lambda, forming a mixed graphon. Finally, graphs are sampled from the mixed graphon as synthetic graphs for training.  The proposed method is evaluated using graph classification accuracy and robustness to graph corruption, showing improvement over baseline models. Applying Mixup to graph samples is an interesting and important research topic, but I think the paper in its current form is a bit immature.

**Summary Of The Review:**

Some key assumptions of the proposed method are not well justified. Also, the experimental study is quite weak in its current form.

---

> ### Author Response · Authors · 2021-11-17
> **Response to Reviewer RkbU ( part 3 / 3 )**
>
> > **3(f).... why one needs extra computation cost for g-Mixup?**
>
> - G-mixup has no labor costs compared to data collection, which makes G-mixup low-cost. In this sense, it is worthing to augment the existing dataset to improve the performance, which is one of the most important goals of data augmentation. Besides, in the updated version of Table 2, our methods still reach a reasonable performance improvement. Tables 2, 5 and 6 show G-Mixup obtain $12/15$, $7/9$, $5/5$ best performance among all the baselines and datasets, which shows the G-Mixup improves the performance substantially without extra cost of data collection.
>
> > **3(g). Also, for dropEdge, 4 layer GCN and 5 Layer GIN seem a bit too shallow to me. DropEdge typically performs well on deeper GNNs such as six to eight layers.**
>
> - Because g-mixup is a plug-in model-agnostic graph data augmentation method, we choose the commonly used graph neural networks (4-layer GCN and 5-Layer GIN) as our backbones.
> - We added experiments to show the impact of the number of layers on the performance of G-Mixup. We concluded that **G-Mixup could improve GCN with various layers**. The additional experiments are presented in **Appendix C.4** The brief results are listed as follows:
>
> | dataset  |  Mehtod | 2 | 3 | 4 | 5 | 6 | 7 | 8 | 9 |
> | :--- |   :---- | :---: | :---:|  :---:  |:---: | :---:|  :---:  |:---: | :---:|
> |  IMDB-BINARY   | vanilla     | 70.33   |**80.91** | 72.18 | 72.16 | 73.00 | 64.33 |  56.00 | 47.66 |
> |                | w/ Dropedge | 72.00   |**77.12** | 72.50 | 72.50 | 74.12 | 59.75 |  55.87 | 49.12 |
> |                | w/ G-Mixup  | 73.25   |**83.25** | 72.87 | 74.50 | 74.62 | 62.20 |  53.37 | 47.37 |
> |  REDDIT-BINARY | vanilla     | **85.50**   |80.91 | 78.82 | 80.41 | 83.00 | 74.50 |  79.66 | 67.08 |
> |                | w/ Dropedge | **82.62**   |77.12 | 81.25 | 76.18 | 78.50 | 73.56 |  76.37 | 73.25 |
> |                | w/ G-Mixup  | 86.93   |83.25 | 89.81 | 87.87 | **90.00** | 85.93 |  83.62 | 74.12 |
>
>
> > **3(h). In Table3, it would be useful to also compare with dropEdge and GraphMixup.**
>
> - We added Dropedge and ManifoldMixup as our baselines in Table 5.  The results show that **G-Mixup is more robust than DropEdge and ManifoldMixup** in the context of label corruption and topology corruption.
>
>
> > **4. “Prohibit” here is a bit strong to me.... Last paragraph in section 1. “Since directly mixing up graphs is intractable,”.... why and how the intractable occurs here?**
>
> - Thanks for your suggestion. We changed the sentence “thus prohibit us from directly adopting the Mixup strategy to graph data” to "thus make it challenging to directly adopt the Mixup strategy to graph data".
> - The reasons for *intractable* are that 1) The nodes are not the same between the graphs, and the connection information is non-Euclidean. 2)graph nodes are not readily ordered or aligned. The traditional mixup is performed as an interpolation between samples in the Euclidean space. However, it is unclear how to handle connections or structures in graph data in non-Euclidean space, as nodes between different graphs are not aligned.  These unique characteristics of graphs make directly mixing up two graphs intractable.
>
> > **5. Notations are inconsistent. Such as N(v_{i}) vs. N(i); k and n refer to nodes and graphs in different places. Typos: first sentence of section 4.2 “propose the following the theorem.”; middle of second paragraph in section 4.3: “on a estimated…”.**
>
> - Thank you for pointing typos out. We modified them in the updated version.
>
> Thank you again for your constructive comments. We are looking forward to your feedback and are glad to answer your follow-up questions.
>
> **[References]**
>
> [1] Xu, Hongteng, et al. "Learning Graphons via Structured Gromov-Wasserstein Barycenters." AAAI 2021.
>
> [2]Ruiz, Luana, Luiz Chamon, and Alejandro Ribeiro. "Graphon neural networks and the transferability of graph neural networks." NeurIPS 2020.
>
> [3] Su, Yi, Raymond KW Wong, and Thomas CM Lee. "Network estimation via graphon with node features." IEEE Transactions on Network Science and Engineering, 2020
>
> [4] Parada-Mayorga, Alejandro, Luana Ruiz, and Alejandro Ribeiro. "Graphon pooling in graph neural networks." IEEE EUSIPCO 2022.
>
> [5] Chatterjee, Sourav. "Matrix estimation by universal singular value thresholding." The Annals of Statistics 2015.
>
> [6] Guo, Hongyu, et al. "Mixup as locally linear out-of-manifold regularization." AAAI 2019.
>
> [7] Wang et al., Mixup for Node and Graph Classification, WWW 2021.

---

> ### Author Response · Authors · 2021-11-17
> **Response to Reviewer RkbU ( part 2 / 3 )**
>
> > **3(a). A critical baseline (MixupGraph) is missed in the experiment.**
>
> - Thanks for your suggestion. We added the MixupGraph[7] as a baseline in our experiments in the updated version. The results show G-Mixup performs better that MixupGraph (ManifoldMixup).
>
> > **3(b)...provide ablation studies to show how accurate are the graphons estimated for the datasets (namely Equation 3), and how such estimation impacts the performance of the g-Mixup.**
>
> - The real-world graph datasets do not have ground-truth graphons, making it hard to evaluate the graphon estimation in real-world scenarios. Typically graphon estimation methods are evaluated on the synthetic graphons[1,6]. The graphon estimation methods adopted in our works are theoretically and experimentally valid to estimate a good enough graphon, as mentioned in Question 1.
>
> > **3(c). There is a parameter n, i.e., the number of nodes to be generate by the graphon. How this number impacts the performance of g-Mixup?**
>
> - We experimented to show the impact of nodes number of synthetic graphs generated by G-mixup, and the additional experiments are presented in **Appendix C.3**.  The results show that the adopting average number of nodes in original graphs is reasonable for this parameter $n$. The number of nodes is the same as the size $K$ of a matrix-form step function $\mathbf{W}$, but it would be various due to the randomness of graph sampling.  We present the results as follows (the average nodes number of IMDB-BINARY and REDDIT-BINARY are 19.77 and 429.63). The results show that the average number of nodes in original graphs is a good choice for the number of nodes to be generated.
>
> | $n/K$   | 10 | 15 | 20 | 25 | 30 | 35 | 40 |
> | :---: |   :----: | :---: | :---:|  :---:  |:---:|  :---:  | :---:  |
> |  IMDB-BINARY   | 73.75 | 74.50 | 74.62 | 72.37| 72.37| 73.87| 72.37|
>
>
> | $n/K$  | 200 | 250 | 300 | 350 | 400 | 450 | 500 | 550 | 600 | 650 | 700 | 750 | 800 |
> | :---: |   :----: | :---: | :---:|  :---:  |:---:|  :---:  | :---:  | :---:  | :---:  | :---:  | :---:  |:---:  | :---:  |
> |  REDDIT-BINARY   | 84.43 | 85.62 | 88.25 | 88.43 | 89.06 | 89.93 | 89.50 | 88.75 | 89.68 | 89.87 | 89.75 | 87.37 | 90.25 |
>
>
> > **3(d). When estimating graphons from training data, K is used as the average number of nodes in all graphs. How the K was set in testing is not clear to me, and how different Ks impact the g-Mixup performance?**
>
> - Hyperparameter $K$ has no impact on the testing time because we do not augment the graph data in the testing time. The graph data augmentation is typically conducted before/in the training phase, and $K$ is the average number of nodes in the training graphs.
> - We conducted an experiment to show the impact of $K$ on the performance of g-mixup. The $K$ is the nodes number of synthetic graphs generated by the graphon, and it is the same as parameter $n$ in Question (3c).
>
> > **3(e). The datasets used are without nature node features, I wonder how the proposed method would perform on graphs with node features.**
>
> - We generate node features of synthetic graphs based on the original two sets of graphs. Specifically, we generate the node feature of each graphons. In the graphon estimation, we align the node features with the process of the adjacency matrix. For each graphon, we have a set of node features. Then we can pool the node features to obtain the *graphon features*. The node features would inherit from the graphon features. We also revise Section 3.2 by adding how to generate the node features.

---

> ### Author Response · Authors · 2021-11-17
> **Response to Reviewer RkbU ( part 1 / 3 )**
>
> We greatly appreciate your constructive comments. We conducted additional experiments to address your primary concern about our experiments, which would improve our work largely. Below we address your concerns detailedly.
>
> > **1.each class has a graphon and this graphon can be accurately estimated. I would like to see some theoretical proof and/or analysis on that. ...**
>
> - "each class has a graphon" is an assumption and basis of g-mixup. Thus we carefully investigate it by experiments.
>   - 1) Figures 2 and 5 (visualization of graphons) empirically show that each class of graphs has different graphons, forming the foundation of g-mixup.
>   - 2) Besides, graphon have been shown its effectiveness in processing real-world dataset[1,2,3,4].
> - "The graphon can be accurately estimated": The adopted graphon estimated methods (e.g., LG, USVT, SBA) are well-studied methods. Typically they have rigorous mathematical proof to upper bound the graphon estimation error. For example, Theorem 2.10 in [5] shows the graphon estimation error of USVT is strictly upper bounded. And we also copy the results of graphon estimation methods on synthetic graphon from [1]. The graphon estimation is based on $10$ graphs, the error is MSE error, and the resolution of graphon is $1000\times 1000$. The results show the graphon estimation methods in our work can precisely estimate graphon.
>
> |  $\mathbf{W}(x,y)$| SBA | LG | MC | USVT | SAS |
> | :--- |   :---- | :---: | :---:|  :---:  |:---: |
> | $xy$ |65.6±6.5| 29.8±5.7| 11.3±0.8| 31.7±2.5| 125.0±1.3|
> | $e^{-(x^{0.7} + y^{0.7})}$ |58.7±7.8| 22.9±3.1| 71.7±0.5| 12.2±1.5| 77.7±0.8|
> | $\frac{x^2+y^2+\sqrt{x}+\sqrt{y} }{}$ |63.4±7.6|  24.1±2.5| 73.2±0.7| 33.8±1.1| 99.3±1.2|
> | $\frac{1}{2}(x+y)$ |66.2±8.3| 24.0±2.5| 71.9±0.6| 40.2±0.8| 108.3±1.0|
> | $\frac{1}{1+exp(-10(x^2+y^2))}$ |55.0±9.5|  23.1±3.2| 64.6±0.5| 37.3±0.6| 73.3±0.7|
>
>
> > **2. at the earlier stage of training, ... how this (Theorem 2) guarantees the proposed g-Mixup would work. ... Also, since the addition and deletion of graph edge in g-Mixup are based on the learned (imperfect) graphon, I think it may encounter the manifold intrusion issue.**
>
> - We generate the synthetic graphs before the training phase and train the models with them. In this way, at the earlier stage of training, we have enough synthetic graphs to reach the requirements of Theorem 2. In our implementation, we treat the G-mixup as preprocessing. We first generate all the synthetic graphs and their label before each epoch since our method is model-agnostic and then train the models with the synthetic graphs and original graphs.
> - G-Mixup does not add or delete edges based on the original graphs, which avoids the manifold intrusion[6] issue to a large extent. Besides, one of the most harmful problems of manifold intrusion in machine learning is that it will lead to the degeneration of the model performance. Our experimental results show that G-Mixup improves the model performance and robustness, indicating that the manifold intrusion may not exist in G-Mixup.

---

> > ### Comment · Reviewer_RkbU · 2021-11-19
> > **Thank you for your feedback to my reviews**
> >
> > I appreciate the authors' feedbacks and the clarification. But I am not fully convinced here.
> >
> > 1. Regarding the assumption of each class has a graphon that can be accurately estimated.
> > I think this is a fundamental assumption to your approach, and indirect empirical observations on some datasets may not be very convincing. Do you suggest that we will first empirically verify this condition through visualization of graphons and then decide if the dataset would be appropriated to use G-Mixup? Besides, what happens to the node and edge features in this  graphon estimation and synthetic graph generation? It is not very clear to me.
> >
> >
> > 2. Regarding the manifold intrusion issue. Since G-Mixup generates synthetic graphs by sampling, how would G-Mixup guarantee that the generated graphs not to collide with the original training graphs or other synthetic graphs? I think the proposed method's  performance evidence is not a direct support to the manifold intrusion free claim. Also, how sensitive is the mixing ratio \lambda to the performance of G-Mixup?

---

> > > ### Author Response · Authors · 2021-11-21
> > > **Response to follow-up questions (part 2 /2)**
> > >
> > > > **2(b). sensitivity of the mixing ratio $\lambda$**
> > >
> > > Thanks very much for your suggestion. We conducted the sensitivity experiments of the mixing ratio $\lambda$ on IMDB-BINARY dataset. The results show that the performance reported in the paper is with $\lambda=[0.15]$ while the performance (in boldface) with best $\lambda$ is much higher than the reported, which suggests that G-Mixup has more potential to improve the performance of graph neural networks.
> > >
> > > | $\lambda$   | 0.05 | 0.10 | [0.15] | 0.20 | 0.25 | 0.30 | 0.35 | 0.40 | 0.45 | 0.50 |
> > > | :---: |   :----: | :---: | :---:|  :---:  |:---:|  :---:  | :---:  | :---:  | :---:  | :---:  |
> > > |  IMDB-BINARY (GCN)   | 71.95±4.37 | 72.93±2.78 | [72.87±3.85] | 73.06±3.63| 72.31±2.12| **74.31±2.21**| 73.06±3.35 |  72.68±3.48 | 73.00±2.20 |  72.75±2.81 |
> > >
> > > | $\lambda$   | 0.05 | 0.10 | [0.15] | 0.20 | 0.25 | 0.30 | 0.35 | 0.40 | 0.45 | 0.50 |
> > > | :--: |   :----: | :---: | :---:|  :---:  |:---:|  :---:  | :---:  | :---:  | :---:  | :---:  |
> > > |  IMDB-BINARY (GIN)   | 71.79±4.94 | 72.75±3.48 | [71.94±3.00] | 72.75±3.17| **73.87±3.00** | 72.68±2.72| 73.37±2.06 | 72.56±1.97 | 72.68±3.93 |  73.18±2.12 |
> > >
> > > Thanks for your timely feedback again. We are looking forward to your feedback.
> > >
> > > **[References]**
> > >
> > > [1] Xu, Hongteng, et al. "Learning Graphons via Structured Gromov-Wasserstein Barycenters." AAAI 2021.
> > >
> > > [2] Su, Yi, Raymond KW Wong, and Thomas CM Lee. "Network estimation via graphon with node features." IEEE Transactions on Network Science and Engineering, 2020

---

> > > ### Author Response · Authors · 2021-11-24
> > > **Response to follow-up questions (part 1 /2)**
> > >
> > > Thanks for your timely reply. Here we address your further concerns.
> > >
> > > > **1(a). Applicability of G-Mixup**
> > >
> > > - Thanks very much for your comments. We argue that we do not need to validate the assumption for unseen datasets before applying G-Mixup since this asumption is widely used and experimentally justified. To make your clear about the jusitification of this asumption, we list the following evidences:
> > >
> > >   - This the assumption is widely used by the previous works and is successfuly utilized to solve real-world problem. For example, experiments on IMDB-MULTI/IMDB-BINARY data in [1] shows that different graphons of each class of graphs can be used to cluster real-world graphs. [2] leverages that similar graphs have same graphon to solve the link prediction problem on real-world friendship network.
> > >
> > >   - The graph dataset used in our work (also widely used for graph classification) are consistent with the assumption. The vislization of graphons (shown in Figrues 2 and 5) experimentally shows the graph datasets we used are consistent with the assumption.
> > >
> > >
> > > > **1(b). Node and edge features in this graphon estimation**
> > >
> > >   - We describe the node features generation step by step.
> > >     - For node feature, we first present a preliminary: In the graphon estimation processing, nodes will be aligned between graphs according to the node degree. Then the node features are generated as follows：
> > >       1. Align node features while aligning the nodes in the graphon estimation
> > >       2. Collect the aligned node features of the set of graphs (their graphon is $W_{\mathcal{G}}$) and average the node features to be the graphon features.
> > >       3. Mix up the graphon features when we mix up the graphon.
> > >       4. Node features of synthetic graphs will be the mixed graphon features when we generate new graphs from mixed graphon
> > >
> > >     - For edge feature, the dataset we used with edge features are ogbg-molhiv,ogbg-molbbbp,ogbg-molbace, and we randomly generated the edge features for them.
> > >
> > >
> > > > **2(a). manifold intrusion free claim**
> > >
> > > - Thanks very much for your comments.
> > >
> > > - We argue that generated graphs by G-Mixup will not collide with the original training graphs or other synthetic graphs with a very high probability. To my understanding, manifold intrusion in graph learning represents that the generated graphs have identical topology but different labels. Here we show that this kind of graph manifold intrusion issue will not happen with a very high probability. Because adjacency matrix $\mathbf{A}\in \mathbb{R}^{K\times K}$ of generated graphs are generated from the matrix-from graphon $\mathbf{W}\in \mathbb{R}^{K\times K}$, we have $\mathbf{A}\_{ij} \stackrel{\text{iid}}{\sim} \text{Bern}(\mathbf{W}\_{ij}), \forall i,j \in [K]$.  On this basis, 1)  the probability of generating two identical graphs from the same graphon $\mathbf{W}$ is $\Pi_{i=1}^{K}\Pi_{j=1}^{K}(\mathbf{W}\_{ij}^2 + (1-\mathbf{W}\_{ij})^2)$, which is extremely small since $0<\mathbf{W}\_{ij}^2 + (1-\mathbf{W}\_{ij})^2< 1$ and $K$ is large enough in the real-world graphs; 2) the probability of generating a new graph that is identical to an original graph (the adjacency matrix is $\tilde{\mathbf{A}}$) is $\Pi_{i=1}^{K}\Pi_{j=1}^{K}(\mathbf{W}\_{ij}^{\tilde{\mathbf{A}}\_{ij}}(1-\mathbf{W}\_{ij})^{1-\tilde{\mathbf{A}}\_{ij}})$, which is extremely small since $0<\mathbf{W}\_{ij}^{\tilde{\mathbf{A}}\_{ij}}(1-\mathbf{W}\_{ij})^{1-\tilde{\mathbf{A}}\_{ij}}< 1$ and $K$ is large enough in the real-world graphs.
> > >
> > >
> > >
> > > - Besides, G-Mixup brings more advantages than disadvantages of manifold intrusion. Based on the experimental results that G-Mixup consistently improves the performance of GNNs, we argue that the harm of manifold intrusion in G-Mixup is acceptable at least, or manifold intrusion does not exist in G-Mixup. Instead, G-Mixup brings more advantages than disadvantages of manifold intrusion. Here we list the advantages of G-Mixup to make you clear as follows,
> > >
> > >   - G-Mixup can preserve the different discriminative motifs of the two sets of original graphs, which is theoretically guaranteed.
> > >
> > >   - G-Mixup can generate synthetic graphs, which is the mixture of the original graphs, shown in Figure 3 by a real-world dataset.
> > >
> > >   - G-Mixup can improve the performance of graph neural networks, even with the possible existence of manifold intrusion.

---

> > > ### Author Response · Authors · 2021-11-24
> > > **Looking forward to any further discussions**
> > >
> > > Dear Reviewer RkbU,
> > >
> > > Thank you for your valuable comments and suggestions again. We are looking forward to any further discussions that would help your re-assessment of our work.
> > >
> > > Sincerely, Authors

---

> > > > ### Comment · Reviewer_RkbU · 2021-11-24
> > > > **Concerns remain**
> > > >
> > > > I appreciate the authors’ for their further clarification. Please see my further comments.
> > > >
> > > > First, I would appreciate that if the authors could stop spamming my email account with the same message.
> > > >
> > > > Second, my concern regarding the very fundamental assumption of this work remains, namely each class in the training data has a graphon that can be accurately estimated. I am not convinced by the authors’ empirical observations as listed. This assumption is so crucial to the proposed strategy that stronger evidences, such as a theoretical guarantee may be needed.
> > > >
> > > > Third, I also have the concern on the proposed work’s effectiveness. The performance improvements as shown in the paper are very minor, and I am not sure where these improvements were really coming from. As shown in previous studies, simply injecting noises into either edge or node features of the graphs can act as an effective model regularizer to improve the predictive accuracy.  In this sense, I am not convinced about the benefits of the proposed method when considering the complicated graph sampling processes, the uncertainty on the graphon estimation, and the extra computation cost needed for the approach.

---

> ### Author Response · Authors · 2021-11-18
> **Looking forward to your feedback**
>
> Dear Reviewer RkbU,
>
> Thank you for your valuable suggestions again. We have responded to your initial comments. We are looking forward to your feedback and are glad to answer your further questions.
>
> Thanks, Authors

---

### Official Review · Reviewer_s4VH · 2021-11-02

**Correctness:** 3
**Technical Novelty And Significance:** 2
**Empirical Novelty And Significance:** 2
**Recommendation:** 3
**Confidence:** 4

**Main Review:**

Pros:
1. The graph-level data augmentation methods have been mainly focusing on augmentation for unsupervised graph learning, and the graph data augmentation for supervised graph classification is rather underexplored. This topic is interesting and worth studying.
2. I like idea of using mixup on graph generators instead of directly on graphs, which is simple yet reasonable.
3. The authors provided good theoretical justification on their proposed method.

Cons:
1. Missing related works: [1][2]
2. The design of G-mixup seems to rely on the assumption of "graph topology between classes are divergent" (point 3 in the end of page 1), the theoratical justification (Thm 1) and observations in Fig 2 both support it. Moreover, other than being able to generate more data to improve the model's generalization, G-mixup also tries to help the model to learn such topological patterns. Therefore, I wonder how much does this information help on the node classification. For example, what if we concate $W_{\mathcal{G}}$ with $\mathbf{h}_G$ for all $G \in \mathcal{G}$ and directly use this new graph representation for node classification? Such ablative experiments would make this work more compelling.
3. In Table 2, I noticed that G-mixup showed significant performance improvement on Reddit-B with GCN, while showing mostly marginal improvements under all other settings. The authors should give more explaination on that.
4. I would recommend the authors to use the term "graph data augmentation" instead of "graph augmentation" to avoid potential confusion with a different problem in graph theory.

[1] Data Augmentation for Graph Neural Networks, AAAI'21\
[2] Graph Contrastive Learning Automated, ICML'21

**Summary Of The Paper:**

This work studied the research problem of graph data augmentation for supervised graph classification. The authors proposed G-mixup that performs mixup on graph data via graphon generators. Due to the irregular characteristics of graphs, instead of directly mixing up the data objects, G-mixup mix up the graph generators.

**Summary Of The Review:**

The proposed method is interesting, but the experiments can be improved.

---

> ### Author Response · Authors · 2021-11-17
> **Response To Reviewer s4VH**
>
> Thank you for your valuable comments. We greatly appreciate that you think the topic is interesting and our proposed method is simple yet reasonable. Below we address your concerns one by one.
>
> > **1.Missing related works: [1][2]**
>
> Thank you very much for your suggestions. We added missing related works in the updated version.
>
> - We added discussion about these two related works[1][2] in Section 6 RELATED WORKS. Related work [1] targets improving semi-supervised node classification task performance and is a model-dependent graph data augmentation method. Related work [2] proposes several simple graph data augmentation methods, which can be used as our baselines.
> - We adopted the simple graph data augmentation methods ( NodeDropping[1], Subgraph[2]) in these papers as our baselines. The updated results are presented in **Table 2 and 5**.
>
> > **2. ...I wonder how much does this information help on the node classification. For example, what if we concate with for all and directly use this new graph representation for node classification? Such ablative experiments would make this work more compelling.**
>
> - Thank you very much for your insightful thoughts, and we think "such topological patterns" would help with the node classification task. From my understanding, the input of node classification typically is one graph. Thus we do not have a set of $\mathbf{W}_{\mathcal{G}},\mathbf{h}_\mathcal{G}$  to obtain a new representation for the graph. Please let me know if I misunderstood your concerns, and we are happy to answer your follow-up questions.
>
> > **3. ..In Table 2, I noticed that G-mixup showed significant performance improvement on Reddit-B with GCN, while showing mostly marginal improvements under all other settings.**
>
> The reason for this phenomenon is two-fold.
>
>   - From the dataset perspective，the divergence of the graphons of different classes in the Reddit-B dataset is more significant than other datasets, which benefits G-mixup because the generated graphs can preserve the different motif from two classes of graphs. Thus it would improve the generalization of GCN. As shown in Figures 2 and 3, the graphons of different classes of Reddit-B are distinct (The first graphon REDDIT-BINARY The graphons of REDDIT-BINARY in Figure 2 shows that graphs of class 0 have one high-degree node while the graphs of class1have two).
>
>   - From the model perspective, the expressive power of GCN is limited, which makes it perform worse on REDDIT-B. The generated graphs will help the training of GCN and improve the generalization of GCN. The results in Figure 5 validate that G-Mixup can improve the generalization of graph neural networks.
>
>
> > **4. '...recommend the authors to use the term "graph data augmentation" instead of "graph augmentation".'**
>
> - Thanks for your valuable suggestion. We have changed the "graph augmentation" to "graph data augmentation" in the revised version of our work, which makes our statement more precise and different from the problem in graph theory (e.g., augmenting paths).
>
> > **5. additional experiments.**
>
> - To address your concerns about the experiments, we conducted additional experiments(**in Section 5.1 and Appendix C**), which includes:
>    - more baselines (NodeDropping[1], SubGraph[1], ManifoldMixup[2]), showing **G-Mxiup achieve the best performance among all the baselines**,
>    - molecular property prediction task (ogbg-molhiv, ogbg-molbbbp,ogbg-molbace), showing **G-Mxiup achieve best performance on this task**,
>    - more graph neural networks (DiffPool, MincutPool, GMT), showing **G-Mxiup can improve the performance of multiple graph neural network backbones**,
>    - experiments on the impact of the nodes number of the generated graphs,
>    - experiments on the performance of GCN with different layers.
>
>
> Thank you again for your constructive comments. We are looking forward to your feedback and are glad to answer your follow-up questions.
>
>
> **[References]**
>
> [1] Zhao et al., Data Augmentation for Graph Neural Networks, AAAI 2021.
>
> [2] You et al., Graph Contrastive Learning with Augmentations, NeurIPS 2020.

---

> ### Author Response · Authors · 2021-11-18
> **Looking forward to your feedback**
>
> Dear Reviewer s4VH,
>
> Thank you for your valuable suggestions again. We have responded to your initial comments. We are looking forward to your feedback and are glad to answer your further questions.
>
> Thanks, Authors

---

> ### Author Response · Authors · 2021-11-19
> **Looking forward to your feedback**
>
> Dear Reviewer s4VH,
>
> Thank you for your valuable suggestions again. We have responded to your initial comments. We are looking forward to your feedback and are glad to answer your further questions.
>
> Thanks, Authors

---

> > ### Comment · Reviewer_s4VH · 2021-11-20
> > **Followup questions**
> >
> > I appreciate the authors' detailed feedbacks and additional experiments. The authors have addressed some of my concerns and here are my followup questions:
> >
> > 1. For my previous concern #2, I'm apologize for my typos. I actually meant graph classification. So my question keeps the same except all "node classification" should be "graph classification".
> >
> > 2. In the newly added experimental results (Appendix C), I noticed that the performance improvements are basically neglectable if considering the standard deviation (especially in Tables 5 and 7). For example, the difference between $76.29\pm0.80$ and $76.24\pm0.98$ seems more likely due to random initialization than model improvement. I found only less than half of the improvements (even over backbones) are *statistically significant* and many baselines are worst than the vanilla backbones. I would appreciate if the authors can provide more detailed and accurate analysis on these results. I cannot agree with claims like: "Observation 9: G-Mixup can significantly improve the performance of GNNs on molecular property prediction task. From Table 7, we observed that the G-Mixup outperforms all the baselines on three molecular datasets"
> >
> > 3. Some of the bolding in the new tables are wrong. For example, DiffPool on Reddit-B (Table 5), GCN-virtual on ogbg-molbace (Table 7). The authors should fix the bolding or correct the numbers.

---

> > > ### Author Response · Authors · 2021-11-21
> > > **Response to follow-up questions( part 2 /2 )**
> > >
> > > - **Experiment 1**: We search the hyperparameter - the mixing ratio $\lambda$ on molecular property prediction task (dataset: ogbg-molbbbp). Compared to the reported performances $[68.45±0.84]，[67.20±1.30],[69.28±1.24],[70.02±1.68]$ in Table 7, the best performances are $69.51±1.20,70.05±1.78,70.17±1.03,70.58±1.55$ with tuned $\lambda$, which improve by $1.54, 4.24,1.28, 0.79$ percent, respectively. We also do the statistical significance test between the G-Mixup and the vinilla model, the p-value is $0.00515,0.0994,0.0109,0.0471$, indicating the $3$($p< 0.05$) improvements are statistically siginificant.
> > >
> > > | $\lambda$   | 0.05 | 0.10 | [0.15] | 0.20 | 0.25 | 0.30 | 0.35 | 0.40 | 0.45 | 0.50 |
> > > | ---: |   :----: | :---: | :---:|  :---:  |:---:|  :---:  | :---:  | :---:  | :---:  | :---:  |
> > > |gcn w/ G-Mixup | 68.23±0.75 | 68.23±1.81 | [68.45±0.84] | 67.54±3.09 | **69.51±1.20** | 67.79±0.82 | 67.60±1.31 | 69.48±2.62 | 67.86±1.02 | 68.78±2.61 |
> > > | gcn-virtual w/ G-Mixup   | 68.57±2.61 | 68.81±1.57 | [67.20±1.30] | 68.64±2.09 | **70.05±1.78** | 68.77±2.31 | 69.11±1.12 | 68.82±0.98 | 69.07±1.48 | 68.37±0.95 |
> > > | gin w/ G-Mixup | 68.20±1.04 | 69.37±70.38 | [69.28±1.24] | 68.89±2.70 | **70.17±1.03** | 66.95±0.92 | 69.86±1.05 | 70.01±1.14 | 68.65±1.03 | 69.73±1.32 |
> > > | gin-virtual w/ G-Mixup | **70.58±1.55** | 69.44±1.88 | [70.02±1.68] | 69.77±0.88 | 69.18±0.87 | 68.17±1.67 | 68.62±1.15 | 69.16±1.87 | 70.15±1.32 | 68.66±0.68 |
> > >
> > >
> > > - **Experiment 2**: We search the hyperparameter - the mixing ratio $\lambda$ on molecular property prediction task (dataset: ogbg-molbace). Compared to the reported performances $[80.73±2.06]，[73.55±4.79],[77.79±3.34],[73.53±3.98]$ in Table 7, the best performances are $80.73±2.06,78.34±1.10,77.79±3.34,79.69±1.37$ with tuned $\lambda$, which improve by $0.00, 6.51, 0.00, 8.37$ percent, respectively. We also do the statistical significance test between the G-Mixup and the vinilla model, the p-value is $0.0227,0.0375,0.0401,0.0427$, indicating the $4$ ($p < 0.05$) improvements are statistically siginificant.
> > >
> > > | $\lambda$   | 0.05 | 0.10 | [0.15] | 0.20 | 0.25 | 0.30 | 0.35 | 0.40 | 0.45 | 0.50 |
> > > | ---: |   :----: | :---: | :---:|  :---:  |:---:|  :---:  | :---:  | :---:  | :---:  | :---:  |
> > > |gcn w/ G-Mixup| 77.41±2.24 | 77.33±2.10 | [**80.73±2.06**] | 78.42±2.25 | 77.98±2.03 | 79.25±1.64 | 75.80±4.31 | 78.40±1.88 | 79.54±1.25 | 77.90±2.67 |
> > > | gcn-virtual w/ G-Mixup | 75.64±4.03 | 76.80±1.74 | [73.55±4.79] | 76.46±1.05 | 73.97±4.11 | 76.55±2.28 | 75.91±2.73 | 77.99±2.59 | **78.34±1.10** | 72.84±5.52 |
> > > |gin w/ G-Mixup | 76.44±2.19 | 75.55±4.05 | [**77.79±3.34**] | 75.20±2.91 | 74.79±2.64 | 76.27±4.61 | 73.02±3.68 | 76.29±3.55 | 75.77±2.30 | 74.12±4.12 |
> > > |gin-virtual w/ G-Mixup  | 74.51±4.91 |74.07 ± 2.76| [73.53±3.98] | 78.85±1.98 | 77.15±2.44 | 76.85±3.42 | **79.69±1.37** | 75.13±5.46 | 77.04±1.37 | 78.63±2.04 |
> > >
> > > 2. To ensure a fair comparison, we set the augmentation and mixup ratios to fixed values $0.15$ for all baseline methods and G-Mixup. Due to the unstable training in graph data (this phenomenon is common in graph classification [1]), we need to run all the experiments ten times and compute the mean for comparison, which will lead to high computational time. Thus we set the baselines and G-Mixup to the same hyperparameters. To address your concerns about the performance of G-Mixup, we summarized the overall performance from each table in our experiments.
> > >   - In Table 2, G-Mixup gains $12$ best performances among $15$ reported accuracies and performs $2.84\%$ better than the vanilla model.
> > >   - In Tables 3 and 4, G-Mixup substantially improves the robustness of graph neural networks to label corruption and topology corruption.
> > >   - In Table 5, G-Mixup gains $7$ best performances among $8$ reported accuracies.
> > >   - In Table 6, G-Mixup outperform vanilla,Dropedge, ManifoldMixup by $1.44\%$, $1.28\%$, $2.01\%$, respectively.
> > >   - In Table 7, G-Mixup can improve the performance of GNNs on molecular property prediction task. Table 7 shows G-Mixup gains $9$ best performances among $12$ reported AUCs.
> > >
> > > > **3. Some of the bolding in the new tables are wrong. For example, DiffPool on Reddit-B (Table 5), GCN-virtual on ogbg-molbace (Table 7). The authors should fix the bolding or correct the numbers.**
> > >
> > > - Thanks very much for pointing the typos out. We have fixed them.  The numbers are correct.
> > >
> > > Thank you again for your constructive follow-up questions. We are looking forward to your feedback and are glad to answer your further questions.
> > >
> > >
> > > **[References]**
> > >
> > > [1] Xu, Keyulu, et al. "How Powerful are Graph Neural Networks?." ICLR. 2018.

---

> > > ### Author Response · Authors · 2021-11-21
> > > **Response to follow-up questions( part 1 /2 )**
> > >
> > > Thanks for your timely feedback. We conducted additional experiments to address your follow-up concerns as follows.
> > >
> > > > **1. For my previous concern #2, I'm apologize for my typos. I actually meant graph classification. So my question keeps the same except all "node classification" should be "graph classification".**
> > >
> > > - Thanks for your clarification. We can not leverage $\mathbf{W}_\mathcal{G}$ for graph classification since we do not have $\mathbf{W}_\mathcal{G}$ for the testing graphs. The reason is that estimation of $\mathbf{W}_\mathcal{G}$ needs the label information because we estimate it using graphs within the same class. But we can not estimate the $\mathbf{W}_\mathcal{G}$ in the testing graphs because testing graphs lack label information.
> > >
> > > > **2(a)...cannot agree with claims like"Observation 9..."**
> > >
> > > - We changed our observation 9 to "G-Mixup can improve the performance of GNNs on molecular property prediction task with the experimental setting for a fair comparison. Table 7 shows that G-Mixup gains $9$ best performances among $12$ reported AUCs."
> > >
> > >
> > > > **2(b). ... the performance improvements are basically neglectable if considering the standard deviation (especially in Tables 5- and 7). .... I would appreciate if the authors can provide more detailed and accurate analysis on these results.**
> > >
> > > - To address your concerns about the improvements of G-Mixup, we list the following analysis of the experimental results.
> > >   1. G-Mixup still has more potential to improve the graph neural network if we tune the hyperparameters for each dataset.
> > >   2. In an entirely fair experimental setting, the overall performance improvements of G-Mixup are significant
> > >
> > >
> > > 1. To further address your concerns about the performance of G-Mixup, we provide new experimental results of G-Mixup with the tuned hyperparameter. The results show that G-Mixup significantly improves graph neural networks' performance while we tune the hyperparameter of G-Mixup. Specifically, we search only one hyperparameter - the mixing ratio $\lambda$ in (${W}_\mathcal{I} = \lambda {W}_\mathcal{G} + (1-\lambda) {W}_\mathcal{H},\lambda \mathbf{ y }_\mathcal{G} + (1-\lambda) \mathbf{y}_\mathcal{H}$) in different dataset with different graph neural network backbones. We list the results of experiments as follows:

---

> > > ### Author Response · Authors · 2021-11-24
> > > **Looking forward to any further discussions**
> > >
> > > Dear Reviewer s4VH,
> > >
> > > Thank you for your valuable comments and suggestions again.  We are looking forward to any further discussions that would help your re-assessment of our work.
> > >
> > > Sincerely, Authors

---

> > > > ### Comment · Reviewer_s4VH · 2021-11-24
> > > > **please stop spamming**
> > > >
> > > > I would appreciate if the authors can respect others' lives and stopping spamming us. You sent the same message to me for ***6 times in 2 days***. Also, deleting your spam comments doesn't make it look good.
> > > >
> > > > As for the paper, I appreciate the authors for the second round of answers and extra experiments. Nevertheless, I still think this work is on (or slightly below) the boarderline and I would like to hear more from the other reviewers before I make my decision on the score.

---

### Author Response · Authors · 2021-11-17
**To All Reviewers**

Dear Reviewers,

Thanks very much for your constructive suggestions and for appreciating the novelty of our work. We respectfully accept your valuable suggestions and added more experiments. We add more results in Tables 2 and 3 in **Section 5** and **Appendix C Additional Experiments For Rebuttal** to report our additional experiments. The major modification of this paper is on **Pages 8, 9, 16, 17, and 18**.  Thank you for your careful reading in advance.

To address your major concerns about the experiments, we try our best to conduct additional experiments, including

  - adding more baselines (ManifoldMixup, NodeDropping, SubGraph) in Section 5.2, showing **G-Mixup achieves the best performance among all the baselines**,

  - adding more graph neural networks (DiffPool, MincutPool, GMT) in Appendix C.1, showing **G-Mixup can improve the performance of multiple graph neural network backbones**,


  - adding molecular property prediction task (ogbg-molhiv, ogbg-molbbbp,ogbg-molbace) in Appendix C.2, showing **G-Mixup achieves the best performance on this task**,

  - adding more baselines (Dropedge and ManifoldMixup) for comparison of robustness, showing **G-Mixup is more robust than DropEdge and ManifoldMixup** in the context of label and topology corruption.

  - adding experiments on the impact of the nodes number of the generated graph in Appendix C.3,

  - adding experiments on the performance of GCN with different layers in Appendix C.4.


Thank you again for your valuable comments and suggestion. We are looking forward to your feedback and are happy to answer your follow-up questions.

---

### Decision · Program_Chairs · 2022-01-20

**Decision:**

Reject

**Comment:**

This work tries to extend mixup to graph structured data, where graphs can differ in the number of nodes, and the space is not Euclidean.  This is achieved by G-Mixup, which interpolates the generator (graphon) of different classes of graphs through the latent Euclidean space.  Experimental results show some promise.

Several concerns have been raised by the reviewers, and although the rebuttal helped, some concerns remain.  For example, how to confirm that the graphon can be accurate estimated.  Several weakness in experiment is also raised, and a revision is needed before the paper can be published.